# Defining the early stages of intestinal colonisation by whipworms

María A. Duque-Correa [1,6✉], David Goulding[1,13], Faye H. Rodgers[1,7,13], J. Andrew Gillis[2], Claire Cormie[1,6], Kate A. Rawlinson [1], Allison J. Bancroft[3], Hayley M. Bennett [1,8], Magda E. Lotkowska[1], Adam J. Reid [1,9], Anneliese O. Speak [1], Paul Scott[1], Nicholas Redshaw[1], Charlotte Tolley [1], Catherine McCarthy[1], Cordelia Brandt[1], Catherine Sharpe[3,10], Caroline Ridley[3,11], Judit Gali Moya[4], Claudia M. Carneiro [5], Tobias Starborg[3,12], Kelly S. Hayes [3], Nancy Holroyd[1], Mandy Sanders[1], David J. Thornton [3], Richard K. Grencis [3,14] & Matthew Berriman [1,14✉]

Whipworms are large metazoan parasites that inhabit multi-intracellular epithelial tunnels in the large intestine of their hosts, causing chronic disease in humans and other mammals. How first-stage larvae invade host epithelia and establish infection remains unclear. Here we investigate early infection events using both *Trichuris muris* infections of mice and murine caecaloids, the first in-vitro system for whipworm infection and organoid model for live helminths. We show that larvae degrade mucus layers to access epithelial cells. In early syncytial tunnels, larvae are completely intracellular, woven through multiple live dividing cells. Using single-cell RNA sequencing of infected mouse caecum, we reveal that progression of infection results in cell damage and an expansion of enterocytes expressing of *Isg15*, potentially instigating the host immune response to the whipworm and tissue repair. Our results unravel intestinal epithelium invasion by whipworms and reveal specific host-parasite interactions that allow the whipworm to establish its multi-intracellular niche.

[1] Wellcome Sanger Institute, Wellcome Genome Campus, Hinxton CB10 1SA, UK. [2] Department of Zoology, University of Cambridge, Cambridge CB2 3EJ, UK. [3] Lydia Becker Institute of Immunology and Inflammation, Wellcome Trust Centre for Cell Matrix Research and Faculty of Biology, Medicine and Health, University of Manchester, Manchester M13 9PT, UK. [4] Faculty of Biology, University of Barcelona, Barcelona 08028, Spain. [5] Immunopathology Laboratory, NUPEB, Federal University of Ouro Preto, Campus Universitario Morro do Cruzeiro, Ouro Preto, MG 35400-000, Brazil. [6] Present address: Cambridge Institute of Therapeutic Immunology and Infectious Disease, University of Cambridge, Cambridge CB2 0AW, UK. [7] Present address: Mogrify Ltd, 25 Cambridge Science Park, Milton Road, Cambridge CB4 0FW, UK. [8] Present address: Genentech, 1 DNA Way, South San Francisco, CA 94080, USA. [9] Present address: Wellcome/Cancer Research UK Gurdon Institute, University of Cambridge, Cambridge CB2 1QN, UK. [10] Present address: InstilBio, UMIC Bio-Incubator, Manchester M13 9XX, UK. [11] Present address: Prime Global Medical Communications, Knutsford WA16 8GP, UK. [12] Present address: Rosalind Franklin Institute, Harwell Campus, Didcot OX11 0FA, UK. [13] These authors contributed equally: David Goulding, Faye H. Rodgers. [14] These authors jointly supervised this work: Richard K. Grencis, Matthew Berriman. ✉email: md19@sanger.ac.uk; mb4@sanger.ac.uk

Human whipworms (*Trichuris trichiura*) infect hundreds of millions of people and cause trichuriasis, a major Neglected Tropical Disease with high chronic morbidity and dire socio-economic consequences in affected countries[1,2]. Although *T. trichiura* is experimentally intractable, a mouse model of infection with the natural rodent-infecting species *T. muris* closely mirrors infections in humans[3], making this species distinctive as the only major soil-transmitted helminth with a direct mouse counterpart.

Whipworms live preferentially in the caecum of their hosts and have a unique life cycle strategy where they establish a multi-intracellular niche within intestinal epithelial cells (IECs)[3–5]. In this niche, whipworms can remain for years causing chronic infections[1,2]. The caecal epithelium is a distinctive tissue; it lacks the villi present in the small intestinal mucosa but, similar to the colonic epithelium, it is composed of a flat epithelial surface (known as the intercrypt table) within which crypts of Lieberkühn are embedded[6–8]. The epithelial composition of the caecum has not been studied at the single-cell level. However, histological studies have shown that it lacks Paneth cells and has numerous goblet cells, although fewer than colonic epithelium[6,7]. The resulting differences in the mucus layers overlying the caecal mucosa and in the microbiota harboured within the caecum[7,9,10], have created a distinct niche in which whipworms have evolved to invade and persist.

Infection with whipworms follows ingestion of eggs from the external environment[1–3]. Upon arrival in the caecum, eggs hatch in a process mediated by the host microbiota[3,11] (Fig. 1a). Within hours, motile first-stage (L1) larvae released from the eggs enter the intestinal epithelia (IE) at the bottom of the crypts of Lieberkühn[4,12–14] (Fig. 1a). To reach this location, L1 larvae need to overcome barriers that are known to protect the crypt base, including the mucus layers covering the epithelium and the continuous stream of fluid that flushes from the crypt to the lumen[15]. To date, the physical and molecular cues directing the larvae to the crypt base and mediating their penetration through the overlying mucus into the IE are unknown.

To accommodate themselves completely inside the IE, the larvae burrow through several IECs creating multinucleated cytoplasmic masses that have been described as syncytial tunnels[16]. The syncytium is thought to provide a sheltered environment and a continuous source of nutrients for the worm, as it moults four times to reach adulthood[5,16–18]. The syncytial tunnels are only readily visible for the first time at the L3 larval stage, around day 21 post infection (p.i.), when the parasites inhabit the epithelia at the top of the crypts and the intercrypt table. The biology and the mechanisms by which the multicellular epithelial burrows are generated upon infection are not understood[1,18]. It is not known whether IECs forming the tunnels are dead, or whether they are alive and interacting with the parasite to orchestrate the development of immune responses[5,16,19,20]. Difficulty in experimentally tracking live L1 larvae over time due to their intractability to genetic manipulation and hence insertion of stable markers, coupled with the fact that they are 100 µm long and embedded in the IECs at the base of the caecal crypts, makes assessing the nature and impact of the early tunnels challenging using conventional approaches[18].

Here, we investigate the cellular context and processes mediating invasion of the IE by whipworm larvae and their establishment within multicellular epithelial tunnels during the first three days of infection. We use a combination of mouse infections, together with an in vitro model comprising murine caecal organoids (caecaloids) infected with *T. muris* L1 larvae, to examine infection biology. Infections of caecaloids recapitulate key early interactions between whipworm larvae and the IE seen in vivo, thus making them the first validated organoid system for infection by a parasitic helminth. We find that L1 larvae degrade the mucus layer and penetrate the underlying IE. We also observe that early syncytial tunnels are composed of mitotically active cells that are alive and actively interacting with the larvae during the first hours of the infection. Progression of infection results in damage to the host epithelium, which responds with an expression signature of type-I Interferon (IFN) signalling dominated by

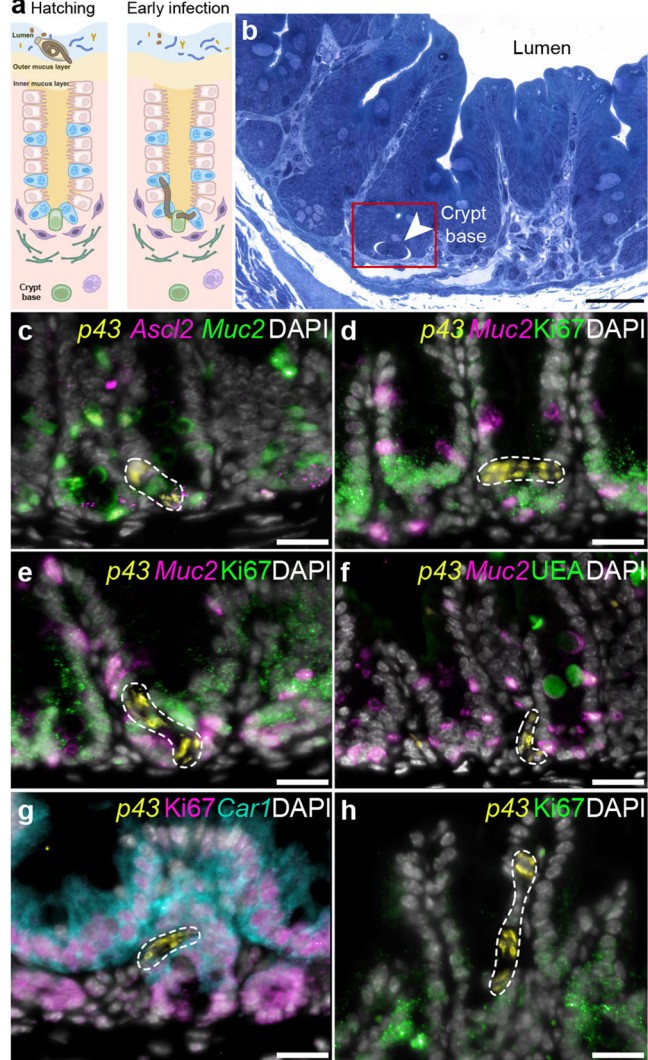

**Fig. 1 Whipworm L1 larvae are predominantly associated with mitotically active cells at the bottom of the crypts of Lieberkühn in the caecum.** **a** Illustration of the processes of egg hatching at the caecal lumen and larvae infection of the IE at the base of the crypts. **b** Representative image of toluidine blue-stained transverse sections from caecum of mice infected with *T. muris* (72 h p.i.), showing whipworm larvae (arrowhead) infecting cells at the base of crypts. Scale bar 30 µm. Image is representative of 15 larvae found during the first 72 h of infection across two independent experiments with three mice per timepoint (3, 24 and 72 h p.i.) each. **c**–**h** *p43*+ *T. muris* larvae (dashed white outline, yellow staining) were detected in close association with: **c** *Ascl2*+ and *Ascl2*+ (magenta)/*Muc2*+ (green) cells (putative stem cells and DSCs, respectively), **d**–**e** Ki-67+ (green)/*Muc2*+ (magenta) DSCs, **f** *Muc2*+ (magenta) DSCs and **g** Ki-67+(magenta)/*Car1*+ (aqua) enterocyte progenitor cells within the dividing zone of the mouse caecal epithelium. **h** In one instance, we observed a larva within the differentiated zone of the caecal crypt. Ki-67+ cells stain in green. In **c** to **h**, nuclei are stained with DAPI (white). Images are representative of 20 larvae found in an experiment with three mice at 72 h p.i. All scale bars = 15 µm.

*Interferon-stimulated gene 15* (*Isg15*), an alarmin that initiates immune and tissue repair responses[21–23]. Our work unravels early host IE-parasite interactions involved in the invasion and colonisation of IE by whipworms, through the formation of syncytial tunnels, that potentially lead to the initiation of host immune responses to the worm.

## Results

**Mitotically active cells at the bottom of crypts of Lieberkühn are the hosts for whipworm L1 larvae.** Light microscopy studies dating back 40–50 years have shown *T. muris* L1 larvae infecting cells at the base of the crypts of Lieberkühn in the first hours of an infection[4,12,13]. Confirming these findings, we found L1 larvae invading IECs in the crypt bases of the caecum of *T. muris*-infected mice as early as three hours p.i. (Fig. 1b; Supplementary Fig. 1a, b). To identify the cell types infected by the L1 larvae, we used mRNA in situ hybridization (ISH) by chain reaction (HCR) to detect the following markers: *Ascl2*, for stem[24,25] and deep secretory cells (DSCs); *Reg4*[26], for DSCs; *Muc2*, for both DSCs and goblet cells; *Car1*, for both enterocyte progenitors and enterocytes; and *Krt20*, for mature enterocytes (Supplementary Fig. 2). We coupled ISH HCR experiments with immuno-fluorescence (IF) to stain Ki-67, which is expressed by mitotically active cells, and stained with the lectin *Ulex europaeus* agglutinin (UEA) to label mucins in DSCs and goblet cells (Supplementary Fig. 2). Transamplifying (TA) cells are defined as Ki-67$^+$ cells that do not express any other of the cell-type markers described above. We found that the caecal epithelium can be broadly divided into a Ki-67$^+$ 'dividing zone', which includes stem, TA, DSCs and enterocyte progenitor cells at the bottom of the crypts, and a Ki-67$^-$ 'differentiated' zone of *Car1*$^+$/*Krt20*$^+$ enterocytes and *Muc2*$^+$/UEA$^+$ goblet cells at the top of the crypts and intercrypt table (Supplementary Fig. 2g).

*T. muris* is detectable in situ by its expression of *p43* (Supplementary Fig. 3), the gene encoding P43 that is the single most abundant protein secreted/excreted by the parasite[27]. By multiplexing ISH by HCR for *T. muris p43* and the markers of the different cellular populations of the caecal epithelia, we were able to locate worms in the caecum of infected mice and test for association with specific host cell types. Nearly all ($n = 19/20$) larvae recovered in the caecum were associated with cells in the "dividing" zone (Fig. 1c–g), with only a single larva recovered within the differentiated zone (Fig. 1h).

Transmission electron microscopy (EM) revealed L1 larvae were in direct contact with the IEC cytoplasm as no cell membrane could be seen between the whipworm cuticle and the cytoplasm (Fig. 2a). We found larvae displaced cellular organelles (Fig. 2) and burrowed through mucin secretory granules of DSCs, possibly causing mucus discharge (Fig. 2b). Tannic acid staining revealed complex carbohydrate in the immediate vicinity surrounding the larvae, both between host cells and in bordering host cell cytoplasm, likely to be secretions from the worm or the result of disrupting DSCs (Fig. 2b, inset). Infected cells reorganized their cytoskeleton around the worm (Fig. 2c, inset). With infection progression, at 24 and 72 h p.i., chromatin was visibly condensed and fragmenting, indicating the onset of apoptosis (Fig. 2d and e), and some host cells were liquified (Fig. 2e, inset I). Our observations were often limited to histological sections with a transverse view through a single slice of the worm within an IEC. Indeed, due to the intricate topography of the multicellular epithelial niche of the larvae, obtaining longitudinal sections of the complete worm inside its niche proved extremely challenging. However, serial block-face SEM allowed us to capture the entire syncytial tunnel formed by L1 larvae (Supplementary Movies 1 and 2) and revealed that by

24 h p.i., larvae were completely intracellular. A typical syncytial tunnel was composed of ~40 IECs, comprising a mixture of DSCs (19%) and stem, TA and progenitor cells (81%) (Supplementary Fig. 1c). Our results suggest that close interactions of cells in the dividing zone with L1 larvae are critical during invasion and colonisation of the IE by whipworms.

**Caecaloids provide an in vitro infection model that reveals the intricate path of early syncytial tunnels.** Although illuminating, serial block-face SEM is technically demanding and is constrained by the need to find the small percentage of infected IECs (<1%) in the total caecal epithelia and by the intracellular location of the L1 larvae at the bottom of crypts. Moreover, the lack of genetic tools to generate fluorescent larvae and of an in vitro culture system (cell lines do not support infection by whipworm L1 larvae) have severely hampered investigations on the early stages of intestinal colonisation by whipworms[18]. Hence, to further examine the processes of invasion and formation of the syncytial tunnels, we developed the first in vitro whipworm infection model using caecaloids[6]. Caecaloids cultured in an open conformation using transwells generated two-dimensional self-organizing structures that recapitulate caecal epithelia cell-type composition and to some extent crypt spatial organisation, albeit in a "flattened" fashion. The structures comprised tight centres of proliferating (Ki-67$^+$) stem (*Ascl2*$^+$), DSC (*Ascl2*$^+$/*Muc2*$^+$), TA cells and enterocyte progenitors (*Car1*$^+$) (Supplementary Figs. 4a and 5a–e) that resembled the dividing zone at base of the crypts in the caecum (Supplementary Fig. 2). These centres were surrounded by areas of non-proliferative differentiated absorptive, goblet, enteroendocrine and tuft cells, with a mucus layer overlaying the IECs; and polarised microvilli (Supplementary Figs. 4b–i and 5f–g). L1 larvae obtained by hatching *T. muris* eggs in the presence of *Escherichia coli*, to simulate microbiota exposure[11], were directly cultured with the IECs. We found L1 larvae infecting IECs in caecaloids as evidenced by enterocyte microvilli (villin) staining above the worm (Fig. 3a) and images showing the larvae woven through multiple IECs within or adjacent to Ki-67$^+$ centres (Fig. 3b). In fact, from a single experiment at 72 h p.i., we recovered the majority of larvae ($n = 27/30$) inside or in direct contact with clusters of Ki-67$^+$ IECs (Fig. 3c). Further experiments co-staining caecaloids for Ki-67 and *Car1* revealed 4/7 and 5/7 recovered larvae associated with cells expressing both markers at 24 and 72 h p.i., respectively; while experiments co-staining for Ki-67 and *Muc2* revealed 6/10 and 5/11 recovered larvae associated with cells expressing both markers at 24 and 72 h p.i., respectively (Fig. 3c). Therefore, the *T. muris*-infected caecaloids effectively replicate the in vivo interactions between L1 larvae and various mitotically active caecal cell types, including DSCs and enterocyte progenitors.

We captured invasion of the mucus layers by L1 larvae with SEM (Fig. 4a), and by TEM found larvae within the cytoplasm of several IECs of the caecaloids (Fig. 4b). Our caecaloid system also revealed the intricate path of the tunnels formed by L1 larvae burrowing through IECs (Fig. 4c, Supplementary Movies 3 and 4). Together, these images show L1 larvae infecting IECs in vitro, effectively reproducing in vivo infection and the interaction of the parasite with the IECs in the dividing zone of the caecal crypts. The caecaloid model enabled the entirety of the L1 larva, its host cells, and the trail of the syncytial tunnels to be visualised.

**Whipworm larvae invade the caecal epithelium by degrading the overlaying mucus layer.** After hatching and to counter host peristalsis, whipworm L1 larvae rapidly reach the bottom of the crypt and invade the IECs[13]. But first, the larvae must traverse the outer and inner mucus layers overlaying the IE. Despite their

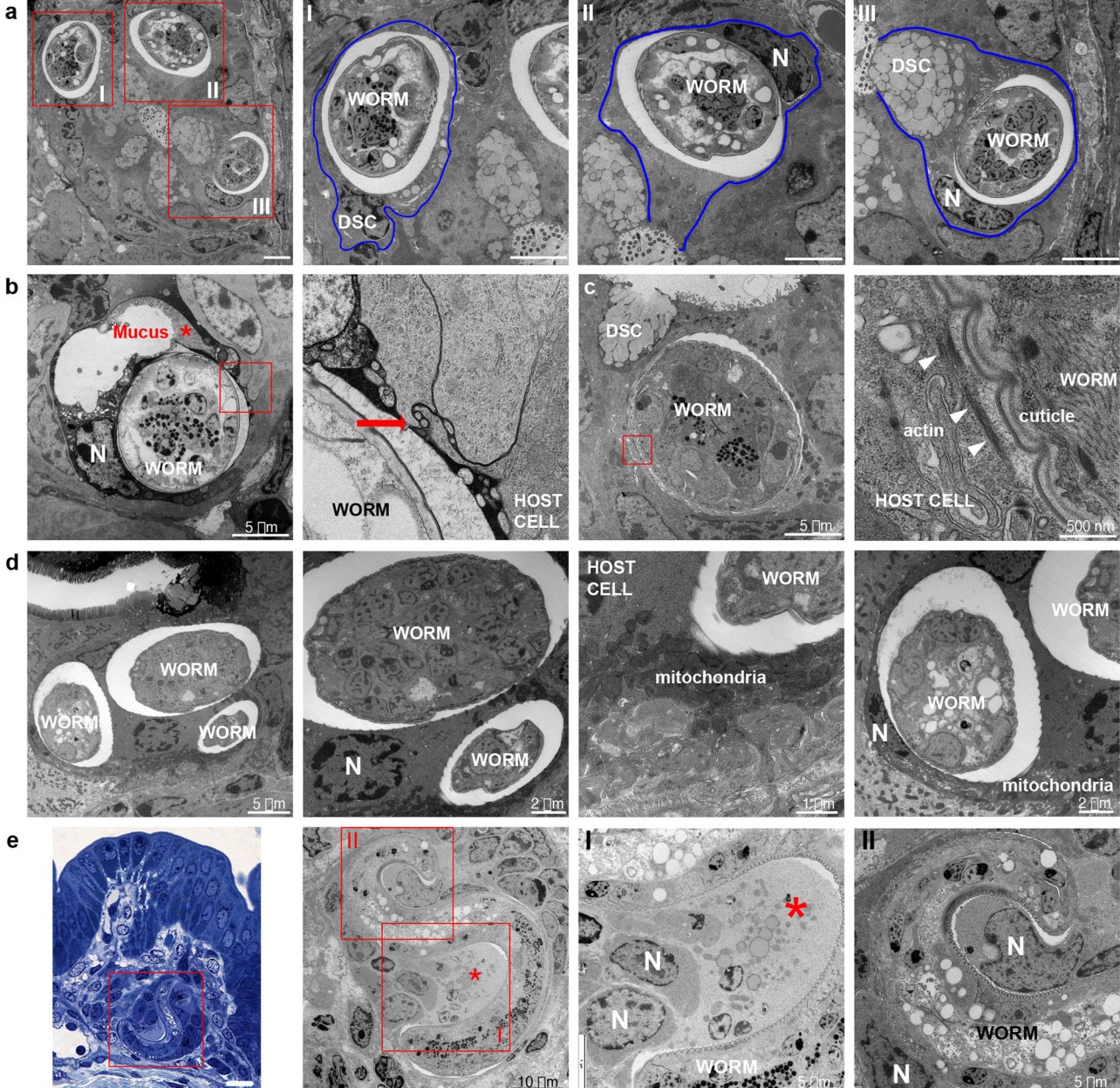

**Fig. 2 Whipworm L1 larvae become completely intracellular and are in direct contact with the cytoplasm of their host cells.** TEM images of transverse sections from caecum of mice infected with *T. muris*. **a** Whipworm L1 larvae infecting DSCs (identified by the mucin secretory granules; insets I and III) and other IECs at the base of the crypt (inset II) at 3 h p.i. Scale bars 5 μm. Blue lines show the cellular membranes of the host cells. **b, c** L1 larvae infecting DSCs in the caecum of mice 3 h p.i., note: **b** potential mucus discharge (red asterisk) and tannic acid staining (black secretion) revealing complex carbohydrate in the host cell cytoplasm and between cells (inset, red arrow); **c** host cell cytoskeleton reorganization of actin filaments adjacent and parallel to the cuticle of the larvae (inset, white arrowheads). **d** L1 larvae infecting several IECs in the caecum of mice at 24 h p.i. Host cells display DNA condensation and fragmentation (pyknotic nuclei, characteristic for the onset of apoptosis) and their nuclei and mitochondria are displaced by the worm. **e** Toluidine blue-stained (scale bar 20 μm) and TEM images of transverse sections from caecum of mice infected with *T. muris*, showing a syncytial tunnel formed by L1 whipworm larvae through IECs (72 h p.i.), and depicting liquefaction of cells (inset I, red asterisk) and nuclei in early stages of apoptosis (inset II). N, nuclei. Images are representative of 15 larvae found during the first 72 h of infection across two independent experiments with three mice per timepoint (3, 24 and 72 h p.i.) each.

motility, mucus can aggregate around whipworm larvae, blocking their advance towards the IE. An additional mechanism is therefore required for larval traversal of the mucus layers. Many intestinal pathogens have evolved enzymes to degrade the mucin oligosaccharides via glycosidases, exposing the mucin peptide backbone to proteases[10]. Hydrolysis of mucins causes disassembly of the polymerized mucin network, reducing mucus viscosity, increasing its porosity and likely impairing mucus barrier function[10,28]. The protozoan parasite *Entamoeba histolytica* breaks down the mucus network to facilitate invasion of IECs by cleaving mucin 2 (MUC2), the major component of intestinal mucus[29]. Adult *T. muris* also degrades MUC2 via secretion of serine proteases[28]. RNA sequencing (RNA-seq) analysis of eggs, L1 and L2 larvae recovered from infected mice showed there are clear groups of serine proteases (Supplementary Fig. 6a), as well as WAP, Kunitz and serpin classes of peptidase inhibitors

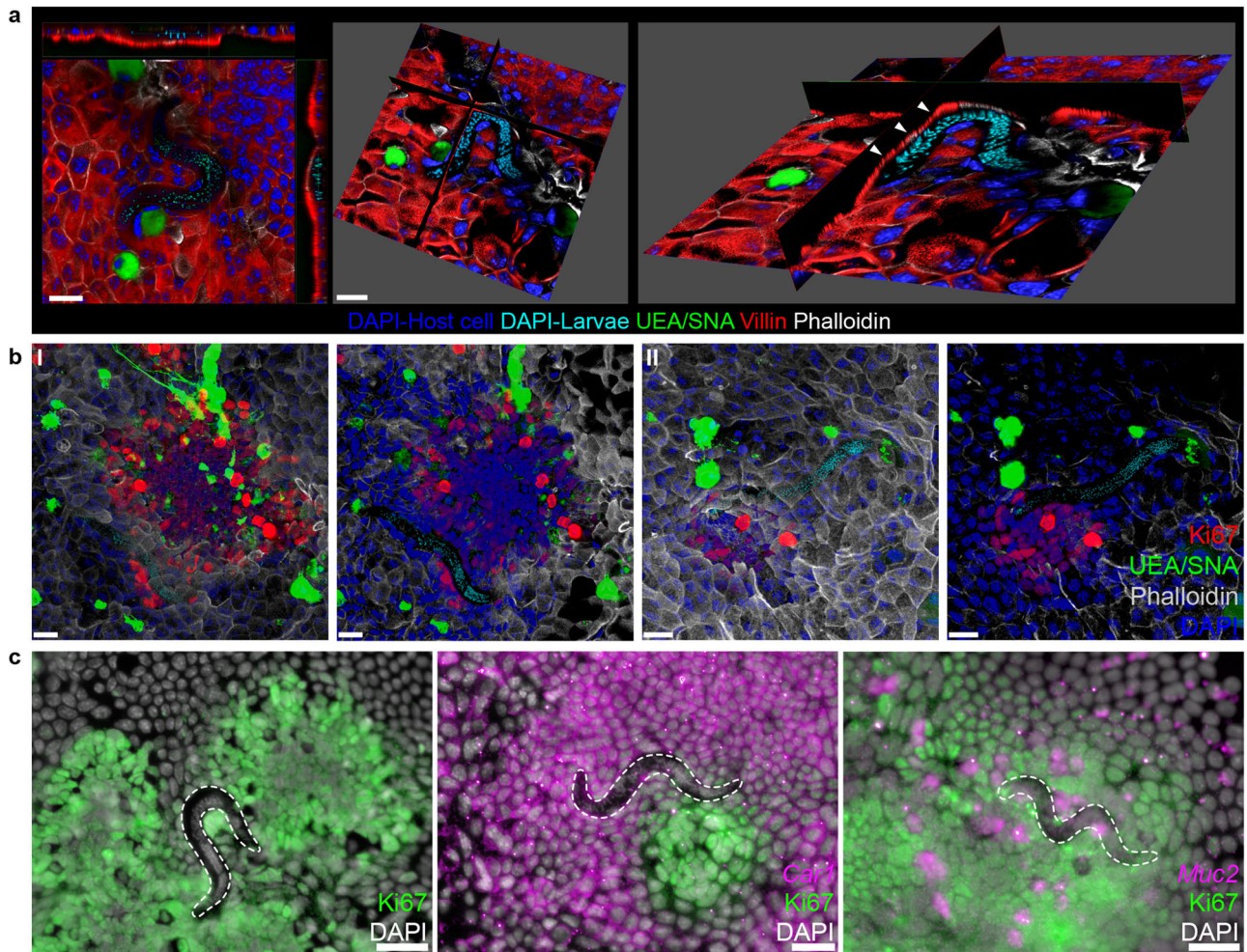

**Fig. 3 Caecaloid—*T. muris* in vitro model reproduces in vivo infection. a, b** Representative confocal immunofluorescence (IF) images of caecaloids infected with whipworm L1 larvae for 24 h. **a** Orthogonal slice visualising enterocyte microvilli (villin staining in red) above the larvae (white arrowheads). Scale bars 20 μm. **b** Complete z-stack projection showing larvae infecting IECs within or adjacent to Ki-67+ (red) dividing centres. In green, the lectins UEA and SNA bind mucins in goblet cells; in blue and aqua, DAPI stains nuclei of IECs and larvae, respectively; and in white, phalloidin binds to F-actin. Scale bars 20 μm. IF imaging experiments on *T. muris*-infected caecaloids were done in triplicate across more than ten independent replicas using three caecaloid lines derived from three C57BL/6 mice. **c** Representative images of *T. muris*-infected caecaloids (72 h p.i) showing L1 larvae infecting cells within or adjacent to Ki-67+ (green) dividing centres, specifically *Car1*+ enterocyte progenitors and *Muc2*+ DSCs (magenta) visualised by IF and mRNA ISH by HCR. In white DAPI stains nuclei. Scale bars 15 μm. HCR imaging experiments on *T. muris*-infected caecaloids were done in triplicate across two independent replicas using two caecaloid lines derived from two C57BL/6 mice.

(Supplementary Fig. 6b) that show the highest expression in L1 larvae recovered at 3 and 24 h p.i. It is therefore likely that L1 larvae secrete proteases to degrade mucus and this facilitates their invasion of the IE. Indeed, the sedimentation profile of purified glycosylated MUC2 was altered by exposure to L1 larvae, with a higher proportion of slower-sedimenting mucins indicating a reduction in size due to mucin depolymerization (Fig. 5a). Early in infections, the small ratio of larvae versus IECs in the caecum dilutes any effects the larvae have on the mucus layer; thus, from in vivo experiments, it is not possible to directly determine whether degradation occurs in the mucus layers overlaying the IECs during invasion by whipworm larvae. We therefore used the caecaloid system to examine the effects of larvae on mucin more closely. Consistent with our results from purified MUC2, when comparing mucus from L1 larvae-infected and uninfected caecaloids, we observed an increased proportion of mucin distribution exhibiting a lower sedimentation rate, indicating increased degradation of mucin polymers (Fig. 5b, Supplementary Fig. 7). Moreover, the mucus layer immediately overlaying larvae-infected IECs was less densely stained by toluidine blue than the mucus overlaying neighbouring

uninfected regions and uninfected caecaloids (Fig. 5c and d), again indicating degradation. Therefore, degradation of mucus by L1 larvae likely enables whipworms to penetrate through the mucus layer and invade the underlying IECs.

**Close interactions between *T. muris* L1 larvae and IECs defining the whipworm niche in the syncytial tunnels.** After crossing the mucus layer, the L1 larva becomes intracellular creating a syncytium, a hallmark of whipworm infections (Figs. 2–4). Only previously described for later larval and adult stages[5,16], syncytial tunnels are suggested to form by lateral burrowing of whipworms through adjacent IECs that join to form a single structure housing the parasite. Presently, the interactions between the host IECs and the L1 larvae and the process of formation of the tunnels are not understood[18]. Tilney et al. previously observed that the syncytium around the anterior end of L3-L4 larvae and adult worms is an inert scaffold of dead cells with a brush border cover[5]. In contrast, at early stages of infection of caecaloids, we found that while cells left behind in the tunnel

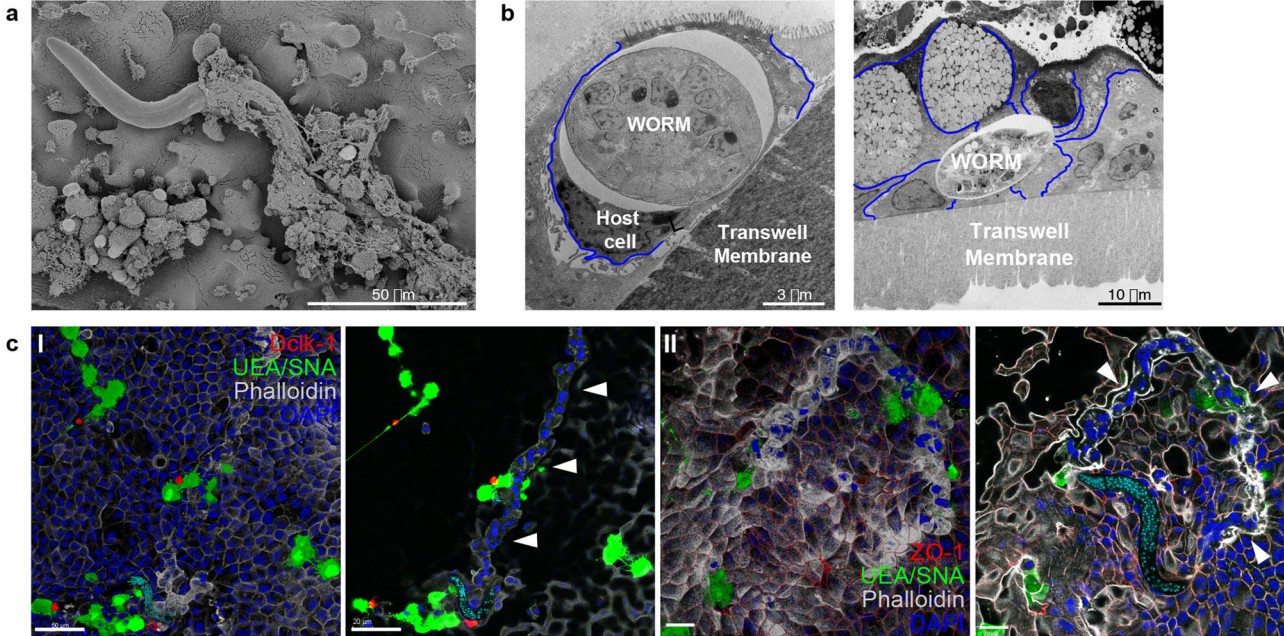

**Fig. 4 Caecaloid—*T. muris* in vitro model reveals intricate path of multi-intracellular tunnels burrowed by whipworm L1 larvae.** Scanning and transmission EM images from caecaloids infected with *T. muris* for 24 h, showing whipworm L1 larvae **a** invading mucus layers and **b** within the cytoplasm of host cells. Blue lines show the cellular membranes of the host cells. **c** Complete z-stack projection and selected and cropped volume of confocal IF images of syncytial tunnels (white arrowheads) in caecaloids infected with L1 whipworm larvae for 24 h. In red, (I) Dclk-1, marker of tuft cells; (II) ZO-1 protein, binding tight junctions; in green, the lectins UEA and SNA bind mucins in goblet cells; in blue and aqua, DAPI stains nuclei of IECs and larvae, respectively; and in white, phalloidin binds to F-actin. Scale bars for (I) 50 µm, and (II) 20 µm. Imaging experiments on *T. muris*-infected caecaloids were performed in triplicate across more than ten independent replicates using three caecaloid lines derived from three C57BL/6 mice.

were dead, the IECs actively infected by the worm were in fact alive (Fig. 6a). Using TEM of caecaloids infected with *T. muris* for 24 and 72 h, we observed direct contact of the L1 larvae with the host cells cytoplasm, displacement by the larvae of cellular organelles (Figs. 4b and 6b–d) and deposition of actin fibres in IECs surrounding the worm cuticle (Fig. 6b, inset I), thus recreating the host-parasite interactions observed in vivo (Fig. 2). With infection progression, at 72 h p.i., we also detected other alterations in infected IECs of caecaloids, including cell liquefaction and pyknotic nuclei indicating early apoptosis (Fig. 6c, d). Moreover, several lysosomes were found in infected cells, many of which were being discharged over the larval cuticle (Fig. 6d insets II and III). Taken together, these findings suggest that early in infection there is an active interplay between the IECs and the parasite at its multi-intracellular niche, which may shape the initial host responses to the larvae.

**Whipworm burrowing through IECs ultimately results in tissue damage.** IE barrier integrity is maintained by intercellular junctions between the IECs including, from apical to basal: tight junctions, adherens junctions, and desmosomes[30]. Syncytial tunnels hosting stage 3 and 4 larvae and adult whipworms present an intact apical surface, stabilised by the actin cytoskeleton and cell junctions, and a basal surface that remains attached to the basement membrane, but lateral membranes of the host IECs are ruptured[5]. In contrast, during early infection, lateral membranes of host cells were still visible and separating their cytoplasm (Figs. 2a and 4b) and tight junctions were still present on infected cells, but had disappeared in the cells left behind in the tunnel as indicated by the presence or absence of ZO-1 stain, in caecaloids (Fig. 4c, II and 7a, Supplementary Movie 5). We noticed that while all intercellular junctions were still present in infected caecaloids cells after 72 h, desmosomes, but not tight and adherens junctions, joining infected IECs and adjacent cells had opened (Fig. 7b, Supplementary Fig. 8a). When

compared to those of uninfected cells, the distance between desmosomes joining infected IECs and adjacent cells significantly increased from 26 nm to 38 nm (Fig. 7c). Strikingly, we observed equivalent perturbations in vivo (Supplementary Fig. 8), further demonstrating that the caecaloid-whipworm model closely recapitulates whipworm infection. Altogether, our results indicate that with progression of infection, the tunnelling of the larvae through the IECs results in IEC damage.

**Host IEC responses to early infection with whipworms are dominated by a type-I IFN signature.** IEC responses to whipworm early infection are thought to initiate host immune responses to the worm and orchestrate repair to the IE damage caused by larval invasion and tunnelling[19,20]. However, currently little is known about the nature of those responses. Using bulk RNA-seq of whole caeca from infected mice at day 7 p.i. and caecal IECs from infected mice at 24 and 72 h p.i., we detected the upregulation of genes involved in innate immune responses, specifically those related to type-I IFN signalling and normally characteristic of bacterial and viral infections (Fig. 8a, Supplementary Fig. 9, Supplementary Data 1 and 2). The response to larval infection in the caecum at day 7 p.i. therefore appeared to reflect the IEC response from a much earlier stage of infection.

Using the 10X Chromium platform, we performed single-cell RNA-seq (scRNA-seq) of caecal IECs from uninfected and *T. muris*-infected mice at 24 and 72 h p.i. Populations of undifferentiated, enterocytes, goblet, enteroendocrine and tuft cells could be identified (Fig. 8b–d, Supplementary Fig. 10). Isolating undifferentiated cells in silico, we further characterised 5 subpopulations: stem and TA cells that are on the S and G2/M phases of cell cycle, DSCs and two enterocyte progenitor populations, which express known markers of these cell types in the small intestine[31,32] and colon[24,25,33–35] (Supplementary Figs. 11 and 12). Enterocytes were divided into five sub-clusters: two early enterocyte populations and three late/mature

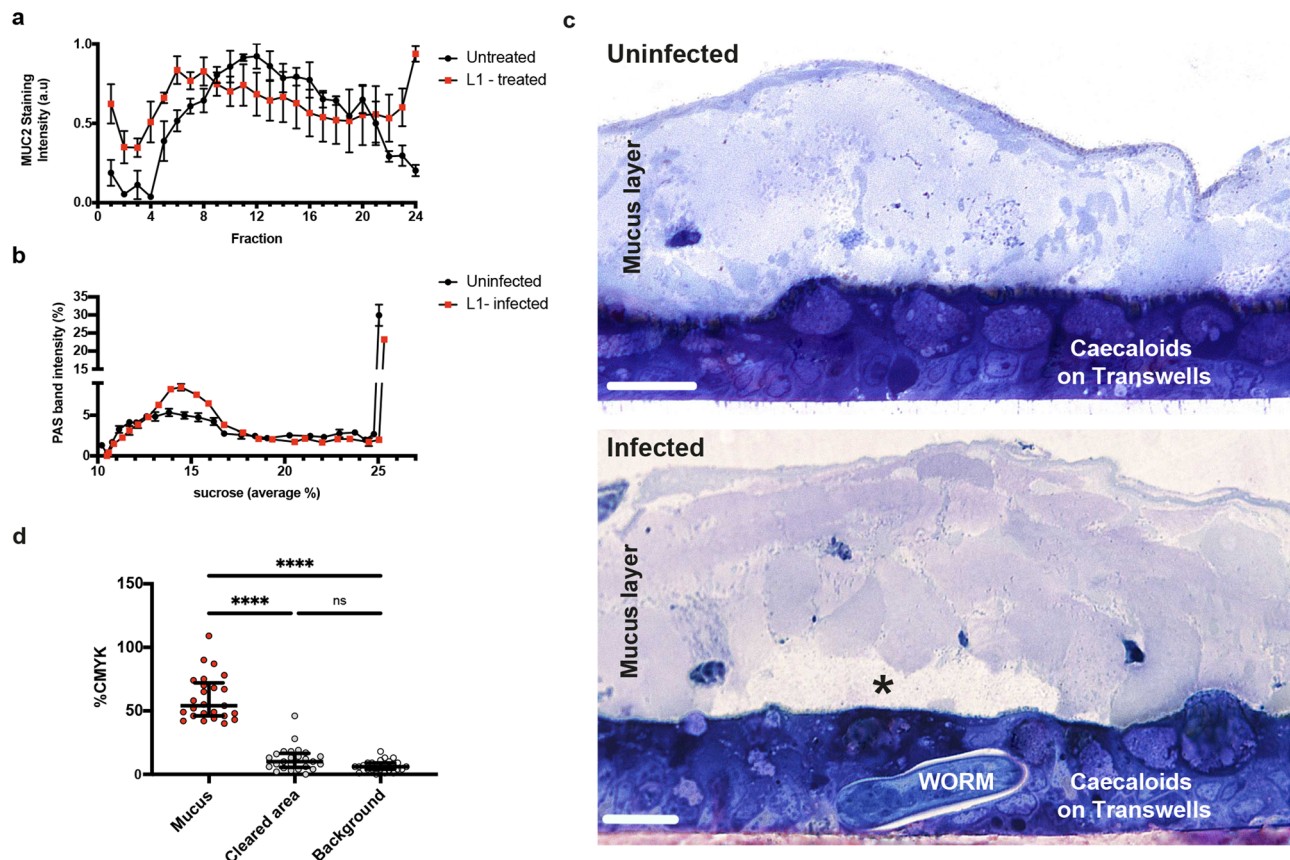

**Fig. 5 Whipworm L1 larvae invade caecal epithelium by degrading the overlaying mucus layer. a** MUC2 purified from LS174T cell lysates was incubated with (red squares) or without (black circles) 400 *T. muris* L1 larvae at 37 °C for 24 h before being subjected to rate zonal centrifugation on linear 6–8 M GuHCl gradients (fraction 1—low GuHCl; fraction 24—high GuHCl). After centrifugation tubes were emptied from the top and the fractions probed with a MUC2 antibody. Data are shown as staining intensity arbitrary units (a.u). Results are represented as the mean +/− standard error of the mean (SEM) of 3 independent experiments. Source data are provided as a Source data file. **b** Caecaloid mucus degradation by *T. muris* L1 larvae at 72 h p.i. Transwells were washed with 0.2 M urea in PBS to recover mucus. Washes were subjected to rate zonal centrifugation on linear 5–25% sucrose gradients. After centrifugation tubes were emptied from the top and the fractions were stained with Periodic Acid Shiff's (PAS) to detect the mucins. Data are shown as percentage of intensity. Black circles represent uninfected caecaloids and red squares represent *T. muris* L1-infected caecaloids. Results are shown as the mean +/− SEM of 3 replicas of one caecaloid line, which are representative of two caecaloid lines. Source data are provided as a Source data file. **c** Representative images of toluidine blue-stained transverse sections from caecaloids uninfected and infected with *T. muris* L1 larvae for 24 h showing degradation (asterisk) of the overlaying mucus layer immediate above the infected cells. Scale bars 20 μm. Imaging experiments on *T. muris*-infected caecaloids were done in triplicate across at least three independent replicas using three caecaloid lines derived from three C57BL/6 mice. **d** Measurement of the density of toluidine blue staining via quantification of the %CMYK recorded (Adobe Photoshop). Five data points from each of three areas were counted: (1) mucus layer overlaying uninfected IECs, (2) cleared mucus above IECs infected with whipworm L1 larvae, and (3) mucus-free background (above and away from the IECs section), for each of five L1 larvae infecting caecaloids. Data are presented as median values with interquartile range. ****$p < 0.0001$ Kruskal Wallis test and Dunn's comparisons among groups. A Source data file is provided.

ones, distinguished by the expression of particular marker genes involved in defined biological processes (Fig. 8b and c, Supplementary Data 3). Interestingly, one cluster of enterocytes ('Entero.Isg15') was characterised by the expression of IFN-stimulated genes (ISGs), specifically *Isg15, Ifit1, Ifitbl1, Ddx60* and *Oasl2* (Fig. 8b and c). We detected a striking increase in the size of this cluster in *T. muris*-infected mice at 72 h p.i. (Fig. 8b and d).

We validated the expansion of Isg15-expressing enterocytes in response to whipworm infection using mRNA ISH by HCR on caecal tissues from uninfected and *T. muris*-infected mice after 24 and 72 h p.i. (Fig. 9a–c). In uninfected mice, occasional crypts expressing high levels of *Isg15* were detected (Fig. 9a), consistent with the presence of this cluster of enterocytes in naive mice (Fig. 8b). These crypts were easily distinguishable above the extremely low baseline level of *Isg15* expressed in enterocytes throughout

the caecum. Interestingly, by 72 h p.i., the number of crypts showing high levels of *Isg15* expression increased significantly, with those *Isg15*+ crypts forming large groups or "islands" (Fig. 9b, c). Using multiplexed ISH by HCR for *T. muris p43*, mouse *Krt20* (mature enterocyte marker, Fig. 8c) and *Isg15*, we were able to locate worms in the caecum of infected mice. Larvae were occasionally found near islands of *Isg15*+ crypts (Fig. 9d; $n = 3/9$ worms detected at 72 h p.i.), though this was not always the case (Fig. 9e; $n = 6/9$ worms detected at 72 h p.i.). We speculate that larval infection and tunnelling through IECs at the bottom of the crypts resulted in the activation of responses by enterocytes immediately above. Taken together, these findings suggest that host IECs responses to early whipworm larvae infection are dominated by a type-I IFN signature driven by the expansion of a distinct population of enterocytes expressing *Isg15*.

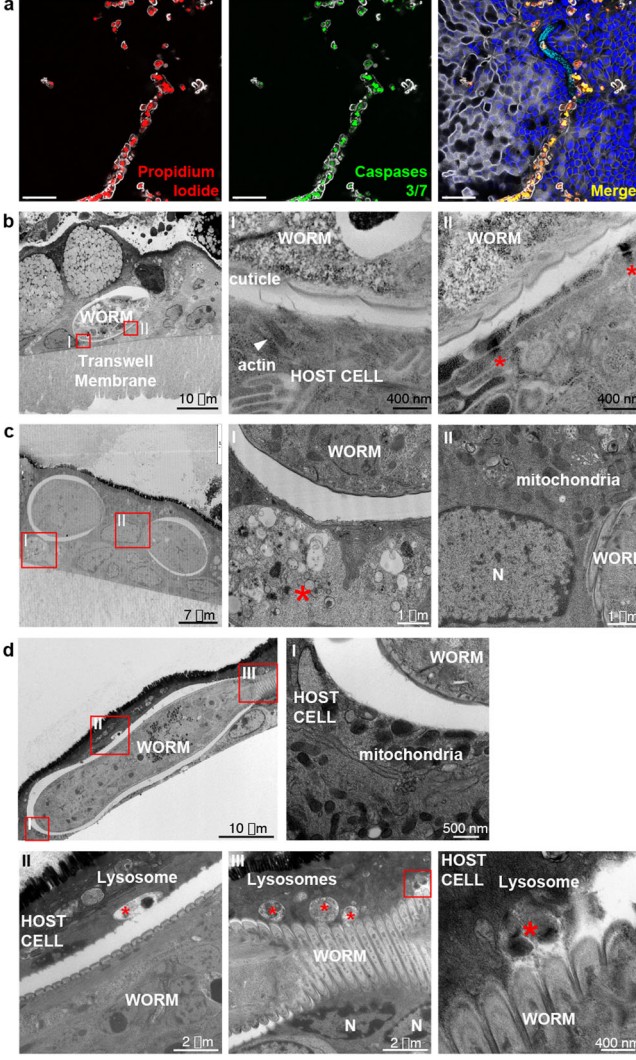

**Fig. 6 Close interactions between *T. muris* whipworm larvae and IECs at syncytial tunnels during early infection of caecaloids. a** Selected confocal IF 2D images from a z-stack showing IECs left behind in the tunnel are necrotic (propidium iodide (red) and caspase-3/7 (green) positive), while IECs infected by worm are alive after 72 h p.i. In blue and aqua, DAPI stains nuclei of IECs and larvae, respectively; and in white, phalloidin binds to F-actin. Scale bars 50 μm. **b**–**d** Representative TEM images of transverse sections of caecaloids infected with *T. muris* L1 larvae, showing host-parasite interactions during early infection: **b** Host cell actin fibres (white arrowhead) surround the cuticle of the worm (inset I) and desmosomes (red asterisks) are still present (inset II) at 24 h p.i. **c** Liquefied cell (inset I, asterisk), and nuclei in early stages of apoptosis (inset II) at 72 h p.i. **d** Displaced mitochondria (inset I) and numerous lysosomes in host cells, some actively discharging over the worm cuticle (insets II and III). N, nuclei; red asterisks, lysosomes. TEM images are representative of 10 larvae during the first 72 h of infection. Imaging experiments on *T. muris*-infected caecaloids were done in triplicate across more than ten independent replicas using three caecaloid lines derived from three C57BL/6 mice.

## Discussion

We have shown that whipworm invasion of the IE is preceded by the degradation of the mucus barrier and we have identified mitotically active cells as the main constituents of the syncytium hosting whipworm L1 larvae. Our findings revealed the early syncytial tunnels are an interactive multi-intracellular niche where whipworm-interplay with host IECs results in the activation of type-I IFN signalling responses that potentially orchestrate the development of immunity against the worm and tissue repair.

Previous studies on syncytial tunnels have focused on L3-L4 larval and adult stages when, on the luminal surface of the caecal and proximal colonic crypts, the syncytium becomes a visible protrusion of host epithelium into the lumen[5,16]. In this work, we determined interactions between *T. muris* L1 larvae and IECs that mediate whipworm penetration of the caecal epithelium and formation of syncytial tunnels. We developed and exploited an in vitro system that used caecaloids, together with in vivo observations where possible. Caecaloids grown and differentiated in transwells recapitulate the complexity of the cellular composition and, to some degree, the organisation of the caecal epithelium. These features of caecaloids promote whipworm larvae infection in vitro enabling us to tackle questions that could not otherwise be investigated in the mouse model.

To our knowledge, our caecaloid-whipworm larvae model is the first example of a live metazoan parasite infecting organoids to establish complex interactions with host cells. While this manuscript was in revision, Smith et al. showed that L3 larvae of *Teladorsagia circumcinta* can invade 3D ovine abomasum and ileum organoids, but by burrowing through Matrigel domes, and passing through the basal side of the epithelium to reach the organoid lumen[36]. In our system, key aspects of the relationship between pathogen and host were maintained. The open conformation of the caecaloids grown in transwells allowed interactions between the L1 larvae and the apical surface of the IE to more closely resemble those occurring in vivo.

Whipworm invasion and colonisation is not supported by epithelial cell lines, 3D caecaloids, or caecaloids grown in transwells that are either undifferentiated (into cell types) or too differentiated (lacking dividing centres). Therefore, a combination of interactions of the larvae with particular cellular or molecular components of the caecal epithelium, as well as specific physicochemical conditions, are critical in triggering parasite invasion. Those components include a complex mucus layer that is well mimicked by our in vitro system, allowing us to visualise mucus degradation in situ and detect depolymerisation of mucins upon larvae infection. Other pathogens that preferentially colonise the large intestine such as *Shigella dysenteriae*[37] and *E. histolytica*[38,39] recognize tissue-specific expression of mucin and mucin glycosylation patterns. *Trichuris* L1 larvae may also recognise molecular cues within the mucus to initiate invasion as Hasnain et al. showed decreased establishment of *T. muris* in *Sat1*[−/−] mice with reduced mucus sulfation[40]. For penetration and tunnelling through IECs, the expression of proteases that we detected in whipworm L1 larvae may be critical because proteases secreted by other parasitic nematode infective larvae, facilitate their entry in the intestinal wall of mammalian hosts[41,42]. Moreover, dividing (Ki-67[+]) centres of caecaloids also replicated the dividing zone at the base of the caecal crypts, and the preferential infection of these cells by *T. muris* L1 larvae suggest the parasite may respond to specific cues from mitotically active cells in order to infect them. Future studies using the caecaloid-whipworm larvae model together with live imaging tools to capture active invasion, and knock-out technologies and inhibitors, will determine the nature of the molecular mechanisms directing larvae movement towards and penetration at the IECs of the crypt base.

Ascertaining the path of the L1 larvae inside the IECs of caecaloids allowed us to discover that, while IECs ultimately succumb after the parasite has moved through them, when the larvae are within the IECs, they remain alive. Surprisingly, during the first 24 h of infection, these host cells present minimal damage. Despite the presence of larva in direct contact with their cytoplasm, which has joined into a multinucleated mass, desmosomal contacts with

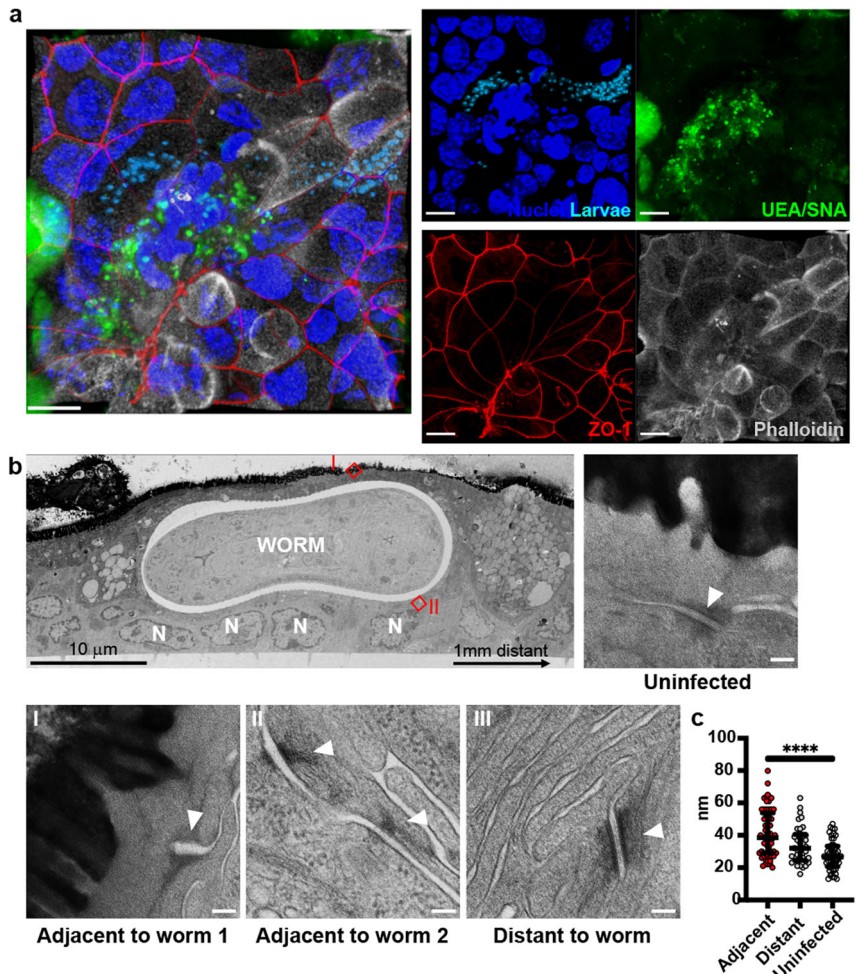

**Fig. 7 Perturbations on desmosomes, but not on tight junctions, in host cells of whipworm larvae during early infection of caecaloids. a** Z-stack projection of confocal IF images of *T. muris* L1 larva in syncytial tunnel in caecaloids infected for 24 h. In blue and aqua, DAPI stains nuclei of IECs and larvae, respectively; in green, the lectins UEA and SNA bind mucins in goblet cells; in red, ZO-1 protein binds tight junctions; and in white, phalloidin binds to F-actin. Scale bars 10 μm. IF imaging experiments on *T. muris*-infected caecaloids were done in triplicate across two independent replicas using three caecaloid lines derived from three C57BL/6 mice. **b** Representative TEM images of transverse sections of *T. muris*-infected caecaloids (72 h p.i.) and desmosomes (arrowheads) joining infected and adjacent cells (insets I and II), cells 1 mm distant to the worm from infected caecaloids (inset III), and cells from uninfected caecaloids. Scale bars for desmosome images 100 nm. **c** Desmosome separation in nm was measured in uninfected cells, cells distant from those infected and host cells from four independent worms. Measurements adjacent $n = 62$, distant $n = 37$ and uninfected $n = 50$. Data are presented as median values with interquartile range. ****$p < 0.0001$ Kruskal Wallis test and Dunn's comparisons among groups. Source data are provided as a Source data file.

adjacent cells and lateral membranes are conserved. We identified distinct host-parasite interactions in the tunnels including the reorganisation of the infected IEC cytoskeleton around the cuticle of the larvae suggesting a response of the host cell to the intracellular parasite. Moreover, we visualised a likely excretion/secretion of products by the larva into its immediate environment in the cytoplasm of the infected IECs. These products may support larval intracellular tunnelling by digesting cells and cellular components, or manipulating the activation of inflammatory responses by the IECs. As infection progressed, at 72 h p.i., we identified infected cells where several lysosomes had surrounded and discharged over the cuticle of the larvae, suggesting a direct response of the host cell to the L1 larvae. In addition, we observed perturbations in IECs forming the syncytial tunnel including opening of the desmosomes, early pyknotic nuclei indicative of apoptosis and liquefaction of cells. These alterations indicate that whipworm burrowing through IECs ultimately results in IE damage.

IE damage at the syncytial tunnels caused by the whipworm larvae either mechanically through tunnelling or chemically as

induced by secretory/excretory products, potentially results in release of damage-associated molecular patterns (DAMPs) and alarmins by infected IECs, which have been suggested to initiate innate immune responses to the worm[19,20,43]. The production of the alarmins IL-25[44,45], IL-33[46,47] and TSLP[48,49] implicated in the induction of type 2 responses to *Trichuris* and other parasitic worms[43] was either not detected or did not significantly increase in our single-cell or bulk transcriptomics data over the first 72 h of infection (Supplementary Figs. 13 and 14). It is remarkable that upon invasion of the epithelium over the very early stages of whipworm colonisation (24 h), IEC responses to the worm are mostly silent. Also, while the parasite infects mitotically active cells, these cells do not seem to respond to *T. muris* infection. Conversely, at 72 h p.i. we identified a striking expansion of an enterocyte population expressing several ISGs including *Isg15*. ISG15 has been described as an alarmin that induces tissue alerts and inflammation via immune cell recruitment, infiltration and activation[21–23]. On the other hand, ISG15 exerts immunoregulatory functions by negatively regulating type-I IFN signalling and production of proinflammatory

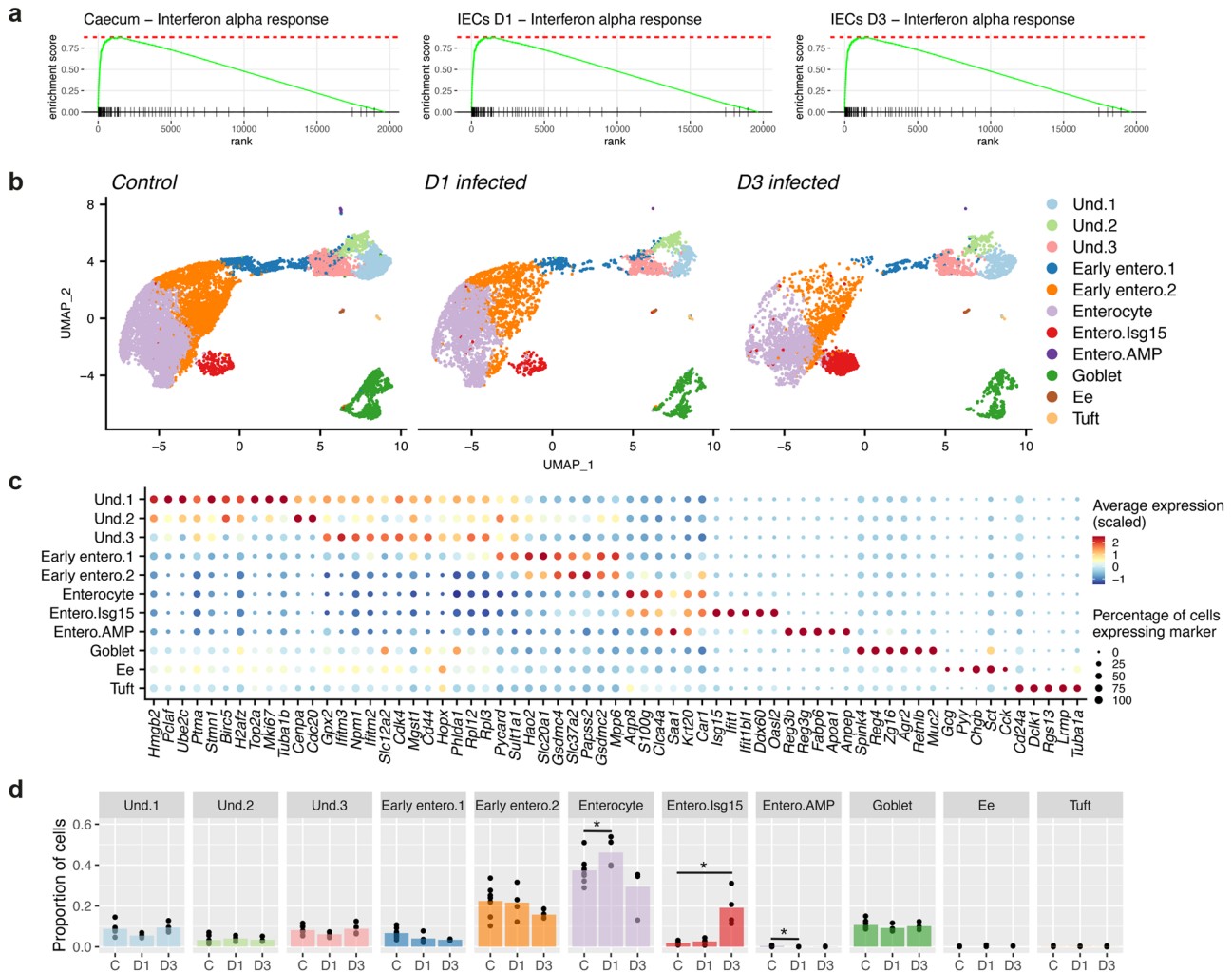

**Fig. 8 Host IECs responses to early infection with whipworms are dominated by a type-I IFN signature. a** Bulk RNA-seq data from complete caecum and caecal IECs of *T. muris*-infected and uninfected mice at days 7, and 1 and 3 p.i., respectively, were analysed by gene set enrichment analysis (GSEA) for cell signature genes in the IFN alpha pathway. All analyses have false discovery rate (FDR) adjusted p-values: Caecum, 0.013; day 1 (D1) IEC, 0.025; day 3 (D3) IEC, 0.026. **b** Uniform manifold approximation and projection (UMAP) plots from single-cell RNA-seq analysis of 22,422 EpCAM$^+$CD45$^-$ cells. IEC populations (colour coded) in the caecum of control ($n = 8$, 4 mice for each timepoint) and *T. muris*-infected mice after 1 and 3 days p.i. ($n = 4$ mice for each timepoint). UMAP representations with separate day-1 and day-3 controls are shown in Supplementary Fig. 10e. **c** Dot plot of the top marker genes for each cell type. The relative size of each dot represents the fraction of cells per cluster that expresses each marker; the colour represents the average (scaled) gene expression. **d** Increased relative abundance of the Enterocyte Isg15 cluster upon 72 h of *T. muris* infection. The size of the clusters, expressed as a proportion of the total number of cells per individual, was compared across four biological replicates at each timepoint for uninfected and *T. muris*-infected mice. Mean +/− standard deviation is shown (*$p = 0.036$ for Enterocyte, *$p = 0.031$ for Entero.Isg15 and *$p = 0.030$ for Entero.AMP, two-tailed *t* test). Source data are provided as a Source data file.

cytokines and chemokines[22,50]. Moreover, ISG15 regulates tissue damage and/or repair responses, specifically in the context of viral infection of the respiratory epithelium[22,51]. Intestinal, respiratory and corneal epithelial cells have been shown to express *Isg15* in response to not only viral, but also protozoal parasitic and fungal infections[22,52–54] and to inflammation during IBD[23]. We therefore hypothesise that ISG15 acts as an alarmin released upon IEC damage caused by whipworm larvae invasion, potentially triggering the development of immune and tissue repair responses.

ISG15 is rapidly induced in response to type-I IFNs but also to nucleic acids sensed by cytosolic receptors in an IFN-independent manner[22]. Our transcriptomic data suggests that the larval induction of *Isg15* in IECs is type-I IFN independent. While we did not detect increased levels of *Ifna1* and *Ifnab1* upon infection either in the complete caecum or IECs (Supplementary Figs. 13 and 14), the expanded *Isg15*-expressing enterocyte population co-expressed

*Ddx60* and *Irf7*. Upon cytosolic-sensing of self- and non-self nucleic acids, DDX60 promotes retinoic-acid inducible gene I (RIG-I)-like receptor-mediated signalling that results in *Isg15* expression via the IFN regulatory factors 3 and 7 (IRF3, IRF7)[22,55]. Thus, DNA or RNA released from whipworm infected/damaged IECs or secreted by larvae may induce *Isg15* expression in infected and bystander cells. Further research is needed to understand the molecular mechanisms leading to *Isg15* expression by IECs and the role of this alarmin in host responses to whipworm infection.

Studies on infection with other parasitic worms, including *Schistosoma mansoni*[56], *Nippostrongylus brasiliensis*[57] and *Heligmosomoides polygyrus*[58] suggest that type-I IFNs are important in driving the initiation of type 2 responses that result in worm expulsion[43]. Our data indicate that this is not the case for *T. muris* but concordant with the establishment of a Type 1 cytokine response that underpins patency and chronic whipworm infection. The lack of activation of

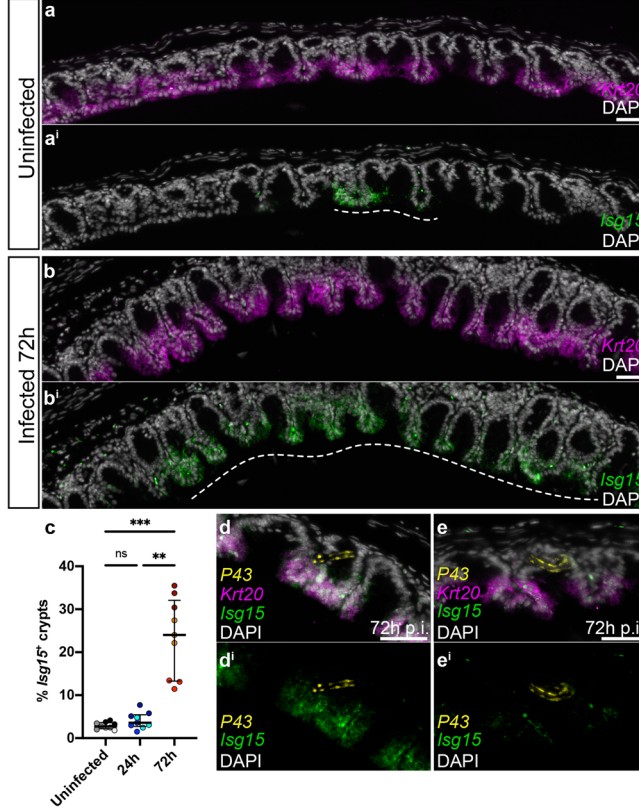

**Fig. 9 Expansion of crypts with enterocytes expressing *Isg15* upon whipworm infection.** Representative images of expression of *Isg15* (green) and *Krt20* (magenta), visualised by mRNA ISH by HCR, in the caecum of **a** uninfected mice and **b** *T. muris*-infected mice after 72 h of infection. Dashed white lines show the extent of "islands" of *Isg15*+ crypts. **c** The number of *Isg15*+ crypts in a caecal section was calculated as a percentage of the total number of crypts across three sections from the same mouse, and with three mice analysed per condition (uninfected, 24 and 72 h p.i.). Data are presented as median values with interquartile range. \*\**p* = 0.0045, \*\*\**p* = 0.0002 Kruskal Wallis test and Dunn's comparisons among groups. For each condition, dots representing technical replicates are coloured identically. Source data are provided as a Source data file. **d** In some instances, worms were located near islands of *Isg15*+ enterocytes, **e** while in other cases, worms were found away from these islands. Scale bars: **a**/**a**^i and **b**/**b**^i = 60 μm; **d**/**d**^i and **e**/**e**^i = 30 μm.

the early canonical damage/alarmin response by a large multicellular pathogen is intriguing. It exemplifies that adaptation to host niche, and subsequent modulation of host protective Type 2 immunity can be achieved by helminths *via* exploitation of divergent early host response pathways.

Collectively, our work has contributed to define the early stages of intestinal invasion and colonisation by whipworms. Extending the applicability of our caecaloid-whipworm system, adapting it to study *T. trichiura* L1 larvae infection of human caecal and proximal colon organoids and adding stroma and immune cells will be important next steps. Further investigations on the early host-parasite interplay within the whipworm mucosal niche will be fundamental to the development of new tools to help control trichuriasis and also provide insights into how the intestinal epithelium adapts to damage and mediates repair.

## Methods

**Mice**. C57BL/6N mice were kept under specific pathogen-free conditions, and colony sentinels tested negative for *Helicobacter* spp. Mice were maintained under a 12-h light/dark cycle at a temperature of 19–24 °C and humidity between 40 and 65%. Mice were fed a regular autoclaved chow diet (LabDiet) and had ad libitum access to food and water. All efforts were made to minimize suffering by considerate housing and husbandry. Animal welfare was assessed routinely for all mice involved. Mice were naive prior to the studies here described. Experiments were performed under the regulation of the UK Animals Scientific Procedures Act 1986 under the Project licenses 80/2596 and P77E8A062 and were approved by the Wellcome Sanger Institute Animal Welfare and Ethical Review Body.

**Parasites and *T. muris* infection**. Infection and maintenance of *T. muris* were conducted as described[59]. Briefly, SCID mice were orally infected under anaesthesia with isoflurane with a high (400) dose of embryonated eggs from *T. muris* E-isolate. Forty-two days later, mice were culled by cervical dislocation and the caecum and proximal colon were removed. The caecum was split and washed in RPMI-1640 plus 500 U/ml penicillin and 500 μg/ml streptomycin (all from Sigma-Aldrich, UK). Worms were removed using fine forceps and cultured for 4 h or overnight in RPMI-1640 plus penicillin/streptomycin (as above) at 37 °C. The E/S from worm culture was centrifuged ($700 \times g$, 10 min, room temperature (RT)) to pellet the eggs. The eggs were allowed to embryonate for at least 6 weeks in distilled water, and infectivity was established by worm burden in SCID mice.

For experiments, age and sex-matched female mice (6–10-week-old) were orally infected under anaesthesia with isoflurane with a high (400–1000) dose of embryonated eggs from *T. muris* E-isolate. Mice were randomised into uninfected and infected groups using the GraphPad Prism randomization tool. Uninfected and infected mice were co-housed. Mice were monitored daily for general condition and weight loss. Mice were culled including concomitant uninfected controls at different time points by cervical dislocation, and caecum and proximal colon were collected for downstream processing. Blinding at the point of measurement was achieved using barcodes. During sample collection, group membership could be seen, however, this stage was completed by technician staff with no knowledge of the experiment objectives.

**In vitro hatching of *T. muris* eggs with *E. coli***. *E. coli* K-12 was grown in Luria Bertani broth overnight at 37 °C and shaking at 200 rpm. Eggs were added to bacterial cultures and incubated for 2 h at 37 °C, 5% $CO_2$. Larvae were washed with PBS three times to remove *E. coli* by centrifugation at $720 \times g$ for 10 min at RT. Bacteria were killed by culturing larvae in RPMI-1640 (Gibco Thermo Fisher Scientific), 10% Foetal Bovine Serum (FBS) (Gibco Thermo Fisher Scientific), 2 mM L-glutamine (Sigma-Aldrich), 1X antibiotic/antimycotic (Sigma-Aldrich) and 1 mg/ml ampicillin (Roche) for 2 h at 37 °C, 5% $CO_2$. Larvae were washed with RPMI-1640 three times to remove ampicillin and separated from egg shells and unembryonated eggs using a stepped 50–60% Percoll (Sigma-Aldrich) gradient. Centrifugation at $300 \times g$ for 15 min at RT was performed and the 50% interface layer was collected. Recovered larvae were washed with RPMI-1640 and resuspended in media containing Primocin (InvivoGen).

**_T. muris_ L1 and L2 larvae recovery from infected mice for RNA extraction**. Mice were culled at 3 and 24 h post infection (p.i.) to recover L1 larvae and at day 14 to recover L2 larvae. Caecum and proximal colon were collected and placed in 5X penicillin/streptomycin (Gibco Thermo Fisher Scientific) in Dulbecco's PBS 1X without calcium and magnesium (PBS) (Gibco Thermo Fisher Scientific). Caecum and proximal were cut longitudinally and were washed to remove faecal contents. The tissues were cut into small sections and added to 0.9% NaCl in PBS and incubated in a water bath at 37 °C for 2 h to allow L1/L2 larvae to come free from the epithelium. Larvae were removed from the NaCl and placed into 1X penicillin/streptomycin PBS. Larvae were washed once with PBS and pellets were resuspended in TRIzol LS (Invitrogen).

**3D caecaloid culture**. Mouse 3D caecaloids lines from C57BL/6 N adult mice (6–8 weeks old) were derived from caecal epithelial crypts as previously described[6]. Briefly, the caecum was cut open longitudinally and luminal contents removed. Tissue was then minced, segments were washed with ice-cold PBS and vigorous shaking to remove mucus, and treated with Gentle Cell Dissociation Reagent (STEMCELL Tech) for 15 min at RT with continuous rocking. Released crypts were collected by centrifugation, washed with ice-cold PBS, resuspended in 200 μl of cold Matrigel (Corning), plated in 6-well tissue culture plates and overlaid with a Wnt-rich medium containing base growth medium (Advanced DMEM/F12 with 2 mM Glutamine, 10 mM HEPES, 1X penicillin/streptomycin, 1X B27 supplement, 1X N2 supplement (all from Gibco Thermo Fisher Scientific)), 50% Wnt3a-conditioned medium (Wnt3a cell line, kindly provided by the Clevers laboratory, Utrecht University, Netherlands), 10% R-spondin1 conditioned medium (293T-HA-Rspo1-Fc cell line, Trevigen), 1 mM N-acetylcysteine (Sigma-Aldrich), 50 ng/ml rmEGF (Gibco Thermo Fisher Scientific), 100 ng/ml rmNoggin (Peprotech), 100 ng/ml rhFGF-10 (Peprotech) and 10 μM Rho kinase (ROCK) inhibitor (Y-27632) dihydrochloride monohydrate (Sigma-Aldrich). Caecaloids were cultured at 37 °C, 5% $CO_2$. The medium was changed every two days and after one week, Wnt3a-conditioned medium was reduced to 30% and penicillin/streptomycin was removed (expansion medium). Expanding caecaloids were passaged, after recovering from Matrigel using ice-cold PBS or Cell Recovery Solution (Corning), by physical dissociation through vigorous pipetting with a p200 pipette every 6 to 7 days.

**Caecaloid culture in 2D conformation using transwells**. 3D caecaloids grown in expansion medium for 4–5 days after passaging were dissociated into single cells by TrypLE Express (Gibco Thermo Fisher Scientific) digestion. 200,000 cells in 200 μl base growth medium were seeded onto 12 mm transwells with polycarbonate porous membranes of 0.4 μm (Corning) pre-coated with 50 mg/ml rat tail collagen I (Gibco Thermo Fisher Scientific). Cells were cultured with expansion medium in the basolateral compartment for two days. Then, basolateral medium was replaced with medium containing 10% Wnt3a-conditioned medium for additional 48 h. To induce differentiation of cultures, medium in the apical compartment was replaced with 50 μl base growth medium and medium in the basolateral compartment with medium containing 2.5% Wnt3A-conditioned medium that was changed every 2 days. Cultures were completely differentiated when cells pumped the media from the apical compartment and cultures looked dry.

**T. muris L1 larvae infection of caecaloids grown in transwells**. Differentiated caecaloid cultures in transwells were infected with 300 L1 T. muris larvae obtained by in vitro hatching of eggs in presence of E. coli. Larvae in a volume of 100 μl of base growth medium were added to the apical compartment of the transwells. Infections were maintained for up to 72 h at 37 °C, 5% $CO_2$.

**IF staining of caecaloids**. For IF, caecaloid cultures in transwells were fixed with 4% Formaldehyde, Methanol-free (Thermo Fisher) in PBS for 20 min at 4 °C, washed three times with PBS and permeabilized with 2% Triton X-100 (Sigma-Aldrich) 5% FBS in PBS for 1 h at RT. Caecaloids were then incubated with primary antibodies α-villin (1:100, Abcam, ab130751), α-Ki-67 (1:250, Abcam, ab16667), α-chromogranin A (1:50, Abcam, ab15160), α-Dcamlk-1 (1:200, Abcam, ab31704), α-zona occludens-1 (ZO-1) protein (1:200, Invitrogen, 61-7300) and the lectins Ulex europaeus agglutinin - Atto488 conjugated (UEA, 1:100, Sigma-Aldrich, 19337) and Sambucus nigra - Fluorescein conjugated (SNA, 1:50, Vector Laboratories, FL-1301) diluted in 0.25% Triton X-100 5% FBS in PBS overnight at 4 °C. After three washes with PBS, caecaloids were stained with secondary antibody Donkey anti-rabbit IgG Alexa Fluor 555 (1:400, Molecular Probes, A31572), phalloidin Alexa Fluor 647 (1:1000, Life Technologies, A22287) and 4',6'-diami-dino-2-phenylindole (DAPI, 1:1000, AppliChem, A1001.0010) at RT for 1 h. Transwell membranes were washed three times with PBS and mounted on slides using ProLong Gold anti-fade reagent (Life Technologies Thermo Fisher Scientific). Confocal microscopy images were taken with a Leica SP8 confocal microscope and processed using the Leica Application Suite X software.

**Cell death fluorescence staining of caecaloids**. To evaluate cell death in infected caecaloids, cultures were incubated with 100 μl warm base growth medium containing 0.3 mg/ml of propidium iodide (Sigma-Aldrich) and 8 μM of CellEvent™ Caspase-3/7 Green Detection Reagent (Invitrogen) for 30 min at 37 °C, 5% $CO_2$. Then, caecaloids were fixed and counterstained as described above.

**mRNA ISH/IF by HCR on paraffin sections of murine caeca and caecaloids**. Caeca from uninfected and T. muris-infected mice after 24 and 72 h p.i. were fixed, embedded in paraffin and sectioned at 8 μm thickness for mRNA in situ hybridization as previously described[60]. All probes, buffers, and hairpins for third-generation HCR were purchased from Molecular Instruments (Los Angeles, California, USA). Lot numbers for the probes used were: Ascl2 (PRI254), Car1 (PRI258), Krt20 (PRG547), Lgr5 (PRB188), Muc2 (PRA885), Reg4 (PRI255) and P43 (PRG548).

mRNA ISH by HCR was carried out on paraffin sections according to the protocol of Choi et al.[61], with modifications according to Criswell and Gillis[62]. Combined mRNA ISH/IF by HCR was carried out on paraffin sections according to the protocol of Schwarzkopf et al.[63]. mRNA ISH and combined mRNA ISH/IF on caecaloids were performed on dissected transwell membranes in a 12-well plate, according to the mammalian cells on a chambered slide protocols of Choi et al.[61] and Schwarzkopf et al.[63], though with volumes scaled up to cover the membrane in the wells.

Immunofluorescent detection of p43 (rabbit anti-p43 (Cambridge Research Biochemicals))[27] on paraffin sections was carried out according to the protocol of Marconi et al.[60], except with heat-mediated antigen retrieval. Antigen retrieval was performed by warming dewaxed and rehydrated slides in water at 60 °C for 5 min, followed by incubation in citrate buffer (10 mM sodium citrate, 0.05% Tween20, pH 6.0) at 95 °C for 25 min. Slides were then cooled for 30 min at −20 °C and rinsed 2 × 5 min in 1X PBS + 0.1% Triton X-100 at RT before proceeding with blocking and antibody incubation. Rabbit anti-p43 primary and AF488-conjugated goat-anti-rabbit IgG secondary (Invitrogen, A11008) antibodies were diluted 1:400 and 1:500, respectively, in 10% sheep serum. All slides were coverslipped with Fluoromount-G containing DAPI (Southern Biotech) and imaged on a Zeiss Axioscope A1 compound microscope.

**Transmission EM**. Caeca and caecaloids were fixed in 2.5% glutaraldehyde/2% paraformaldehyde in 0.1 M sodium cacodylate buffer, post-fixed with 1% osmium tetroxide and mordanted with 1% tannic acid, followed by dehydration through an ethanol series (contrasting with uranyl acetate at the 30% stage) and embedding with an Epoxy Resin Kit (all from Sigma-Aldrich). Semi-thin 0.5 μm sections were

cut and collected onto clean glass slides and dried at 60 °C before staining with 1% Toluidine Blue and 1% Borax (all from Sigma-Aldrich) in distilled water for 30 s. Sections were then rinsed in distilled water and mounted in DPX (Sigma-Aldrich) and coverslipped. Sections were imaged on a Zeiss 200 M Axiovert microscope.

Ultrathin sections, cut on a Leica UC6 ultramicrotome, were contrasted with uranyl acetate and lead nitrate, and images recorded on a FEI 120 kV Spirit Biotwin microscope on an F416 Tietz CCD camera.

**Scanning EM**. Caecaloids were fixed with 2.5% glutaraldehyde and 4% paraformaldehyde in 0.01 M PBS at 4 °C for 1 h, rinsed thoroughly in 0.1 M sodium cacodylate buffer three times, and fixed again in 1% buffered osmium tetroxide for 3 h at RT. To improve conductivity, using the protocol devised by Malick and Wilson[64], the samples were then impregnated with 1% aqueous thiocarbohydrazide and osmium tetroxide layers, with the steps separated by sodium cacodylate washes. They were then dehydrated using an ethanol series (20-min soaks in 30, 50, 70 and 90%, followed by three 20-min washes in 100% ethanol), before they were critical point dried in a Leica EM CPD300 and mounted on aluminium stubs with conducting araldite and sputter coated with a 2 nm platinum layer in a Leica EM ACE 600. Images were taken on a HITACHI SU8030.

**Serial block-face-scanning EM**. Samples from caeca of infected mice were processed according to Deerinck et al.[65]. Embedded tissues were mounted and serial sectioned on a Gatan 3View System and simultaneously imaged on a Zeiss Merlin SEM. Serial images were oriented and assimilated into corrected z-stacks using IMOD. The phenotype of infected cells was determined in each image and the number of cells of each cellular population were quantified.

**MUC2 and caecaloid mucus degradation experiments**. Glycosylated MUC2 was purified from LS174T cells by isopycnic caesium chloride (CsCl) density gradient centrifugation in a Beckman Optima L-90K Ultracentrifuge (Beckman Ti70 rotor) at $117,524 \times g$ for 65 h at 15 °C. Two gradients were performed, the first gradient was run in CsCl/4 M guanidinium hydrochloride (GuHCl) at a starting density of 1.4 g/ml and the second gradient was in CsCl/0.2 M GuHCl at a starting density of 1.5 g/ml[28]. Purified glycosylated MUC2 was incubated with 400 T. muris L1 larvae for 24 h at 37 °C, 5% $CO_2$.

For experiments with caecaloid cultures, after 24 and 72 h of L1 larvae infection, mucus was recovered by six PBS washes followed by six washes with 0.2 M urea.

**Rate zonal centrifugation**. Mucus degradation analysis was conducted as described[28]. Briefly, purified MUC2 (in 4 M GuHCl) was loaded onto the top of 6–8 M GuHCl gradients and centrifuged in a Beckman Optima L-90K Ultracentrifuge (Beckman SW40 rotor) at $201,687 \times g$ for 2.75 h 15 °C. Alternatively, caecaloid mucus samples (in PBS or 0.2 M urea) were loaded onto 5–25% (w/v) linear sucrose gradients and centrifuged in a Beckman Optima L-90K Ultracentrifuge (Beckman SW40 rotor) at $201,687 \times g$ for 3 h 15 °C. After centrifugation tubes were emptied from the top and the fractions probed with a rabbit anti-MUC2 antibody[66–68] (1:1000) or using the Periodic Acid Shiff's (PAS) assay. To determine the sucrose or GuHCl concentration the refractive index of each fraction was measured using a refractometer; all sucrose and GuHCl gradients were comparable.

**Caecal IECs isolation**. Caeca of uninfected and infected mice at day one and three p.i. were processed individually in parallel. Caeca were opened longitudinally, washed with ice-cold HBSS 1X (Gibco Thermo Fisher Scientific) containing 1X penicillin/streptomycin to remove the caecal contents and cut in small fragments. These were incubated at 37 °C in DMEM High Glucose (Gibco Thermo Fisher Scientific), 20% FBS, 2% Luria Broth, 1X penicillin/streptomycin, 100 μg/ml gentamicin (Sigma-Aldrich), 10 μM ROCK inhibitor and 0.5 mg/ml Dispase II (Sigma-Aldrich) with horizontal shaking for 90 min to detach epithelial crypts. Supernatant containing crypts was filtered through a 300 μm cell strainer (PluriSelect) and pelleted by centrifugation at $150 \times g$ for 5 min at RT. Crypts were dissociated into single cells by TrypLE Express digestion 10–20 min at 37 °C. The epithelial single-cell suspension was washed and counted using MOXI automated cell counter. Cells were stained using the antibodies anti-CD236 (epithelial cell adhesion molecule (EPCAM); PE-Cy7, 1:300, Biolegend, 118216) and anti-CD45 (Alexa 700, 1:300, Biolegend, 103128) for 20 min on ice. Cells were washed and stained with DAPI. Live epithelial cells (CD236+, CD45−, DAPI-) were sorted using a fluorescence-activated cell sorting (FACS) Aria flow cytometer (BD) into TRIzol LS for bulk-RNA-seq or DMEM High Glucose 10% FBS 10 μM ROCK inhibitor for droplet-based single-cell RNA-seq.

**Bulk RNA isolation from L1 and L2 T. muris larvae and library preparation for RNA-seq**. L1/L2 larvae were collected via pipette in 200–300 μl of PBS then added directly to lysing matrix D (1.4-mm ceramic spheres, MagNA Lyser Green Bead tubes, Roche) along with 1 ml of TRIzol LS. A PreCellys 24 (Bertin) was used to homogenize the samples by bead beating 3× for 20 s at 6000 rpm, placing the samples on wet ice to cool between runs. Homogenized samples were transferred to 1.5 ml microfuge tubes and total RNA extracted by adding 400 μl of Chloroform, shaking vigorously and incubating for 5 min at RT. Samples were centrifuged at

12,000 × *g* for 15 min at RT, then the upper aqueous phase was transferred to a new microfuge tube prior to addition of 800 μl Isopropyl alcohol with 2 μl GlycoBlue (Invitrogen). Tubes were placed at −80 °C overnight and then centrifuged at 12,000 × *g* for 10 min at 4 °C. The supernatant was removed, being careful not to disturb the blue pellet, and the pellet washed in 1 ml 75% ethanol. The pellet was air-dried briefly then resuspended in nuclease-free water. Total RNA was quantified by Bioanalyzer (Agilent). Multiplexed cDNA libraries were generated from 300 pg of total high-quality RNA according to the SmartSeq2 protocol by Picelli et al.[69], and 125 bp paired-end reads were generated on an Illumina HiSeq according to the manufacturer's standard sequencing protocol.

**Bulk RNA isolation from caecum of uninfected and *T. muris*-infected mice and library preparation for RNA-seq.** Total RNA from sections of caecum of mice pre-infection, and 4 and 7 days p.i., was isolated using 1 ml TRIzol and lysing matrix D to homogenise tissues with a Fastprep24 (MP Biomedicals), and following manufacturer's standard extraction protocol. Total RNA was quantified by Bioanalyzer and 1 μg or 50 μl cherry picked. Poly A mRNA was purified from total RNA using oligodT magnetic beads and strand-specific indexed libraries were prepared using the Illumina's TruSeq Stranded mRNA Sample Prep Kit followed by ten cycles of amplification using KAPA HiFi DNA polymerase (KAPA Biosystems). Libraries were quantified and pooled based on a post-PCR Bioanalyzer and 75 bp paired-end reads were generated on the Illumina HiSeq 2500 according to the manufacturer's standard sequencing protocol.

**Bulk RNA isolation from sorted IECs from uninfected and *T. muris*-infected mice and library preparation for RNA-seq.** Total RNA from sorted IECs 1 and 3 days p.i. with time-matched uninfected concomitant controls, was isolated using Trizol LS. Briefly, 200 μl of Chloroform was added to 500 μl of samples in Trizol LS, shaken vigorously and incubated for 5 min at RT. Samples were centrifuged at 15,000 × *g* for 15 min at RT and the upper aqueous phase was recovered and mixed with one volume of 100% ethanol. RNA was recovered using the RNA Clean and Concentrator kit (Zymo Research). The samples were quantified with the QuantiFluor RNA system (Promega) and 100 ng/50 μl cherry picked. Libraries were then constructed using the NEB Ultra II RNA custom kit (New England BioLabs) on an Agilent Bravo WS automation system followed by 14 cycles of PCR using KAPA HiFI Hot Start polymerase (KAPA Biosystems). The libraries were then pooled in equimolar amounts and 75 bp paired-end reads were generated on the Illumina HiSeq 4000 according to the manufacturers standard sequencing protocol.

**Droplet-based single-cell RNA sequencing 10X.** Sorted cells were counted using a MOXI automated cell counter and loaded onto the 10X Chromium Single Cell Platform (10X Genomics) at a concentration of 1000 cells per μl (Chromium Single Cell 3' Reagent kit v.3) as described in the manufacturer's protocol (10X User Guide). Generation of gel beads in emulsion (GEMs), barcoding, GEM-RT cleanup, complementary DNA amplification and library construction were all performed as per the manufacturer's protocol. Individual sample quality was checked using a Bioanalyzer Tapestation (Agilent). Qubit was used for library quantification before pooling. The final library pool was sequenced on the Illumina HiSeq 4000 instrument using 50 bp paired-end reads.

**Quantification and statistical analysis**

*General.* Desmosome separation measurements in uninfected, distant to worm and adjacent (infected) cells in caecaloids and percentages of *Isg15*+ crypts in uninfected and infected mice after 24 and 72 h p.i. were compared using Kruskal Wallis and Dunn's comparison tests from the Prism 9 software (GraphPad). Statistical comparison for desmosome separation in uninfected and infected mice and the in vitro and in vivo models was performed using Mann–Whitney U two-tailed tests from the Prism 9 software (GraphPad).

*L1 and L2 larvae RNA-seq data processing and analysis.* *T. muris* reference genome (PRJEB126) was downloaded from Wormbase Parasite (v14)[70]. Reads from each RNA-seq sample were mapped against predicted transcripts using Kallisto (v0.42.3)[71]. Indexing and quantification were performed using default parameters. Read counts from multiple transcripts were combined at the gene level. Genes with zero counts across all samples and samples with <500,000 read counts were removed prior to analysis. Differentially expressed genes were determined using DESeq2 (v1.22.2)[72]. Contrasts were performed between egg samples and 3 h L1s, 3 h L1s and 24 h L1s, and 24 h L1s and L2s. Genes were identified as differentially expressed with an adjusted *p*-value of <0.05 and a log2 fold change >1 or <−1. To identify functional patterns in an unbiased way, GO terms were determined which were enriched in differentially expressed genes using TopGO (v2.34.0)[73] (node size = 5, method = weight01, FDR = 0.05, statistic = Fisher).

Gene expression heatmaps were created by combining read counts across biological replicates and calculating log2(FPKM + 1) for each gene, in each condition. These data were plotted using pheatmap (v1.0.11) in R (v3.6.1) for genes which were differentially expressed in any contrast and had the GO term GO:0004252 (serine-type endopeptidase activity), contained a WAP domain (Pfam: PF00095), a Kunitz domain (Pfam: PF00014) or a serpin domain (Pfam: PF00079).

*Caecum and IEC bulk RNA-seq data processing and analysis.* For bulk RNA-seq data from caecum and IEC samples, reads were pseudo-aligned to the mouse transcriptome (Ensembl release 98) using Kallisto (v0.46.2)[71]. The DESeq2 package (v1.26.0)[72] was used to identify differentially expressed genes over the time course (caecum; likelihood ratio test) or by pairwise comparison with time-matched controls (IECs; Wald test). All differentially expressed genes with an FDR adjusted *p*-value <0.05 are reported in Supplementary Data 1 (caecum) and 2 (IECs). Significantly enriched GO terms (Biological Process) annotated to the differentially expressed genes were identified using the GOseq package (v1.38.0)[74], accounting for gene length bias, and *p*-values FDR-corrected with the Benjamini–Hochberg method. Expressed genes were ranked by log(p-value) (most significantly upregulated to most significantly downregulated) and the fgsea package (v1.12.0)[75] used for pathway enrichment analysis with the MSigDB *M. musculus* Hallmark pathways[76].

**10x single-cell RNA-seq analysis**

*Quality control.* Raw count matrices were generated using the CellRanger software suite (v3.0.2), aligning against the mm10-3.0.0 (Ensembl 93) mouse reference transcriptome. Unless otherwise stated, all further analysis was performed using the Seurat package (v3.0.2)[77] in R (v3.6.1). Each library was examined for its distribution of unique molecular identifiers (UMIs) per cell (Supplementary Fig. 7a). UMI filtering thresholds were set for each library at the first local minimum on a UMI density plot, with GEMs associated with fewer UMIs being excluded from further analysis. Cells with a high percentage (>30%) of UMIs associated with mitochondrial genes were also excluded (Supplementary Fig. 7b and c). All libraries were initially merged and clustered prior to doublet detection with the DoubletDecon package (v1.1.4)[78]. The Main_Doublet_Decon function was run 10 times, and the intersection of identified cells were considered to be doublets and removed from the dataset. A distinct cluster of contaminating immune cells (identified by expression of CD3 and CD45) was also removed. Post QC, average cell recovery was 1401 cells per sample, with a total of 22,422 cells captured at a mean depth of 39,336 UMIs per cell and 4910 mean genes per cell.

*Clustering, cell-type identification and pseudotime analysis.* Libraries derived from all mice (four per condition) were merged. Feature counts were log-normalized and scaled, and the 2000 most highly variable genes (identified with the FindVariableFeatures function) used for dimensionality reduction with PCA. The first 35 principal components were used for running UMAP for visualisation (min.dist = 0.05) and for neighbour finding, and clusters initially identified with the FindClusters function (resolution = 0.6). We confirmed the absence of batch effects visually (Supplementary Fig. 7d). Cluster specific marker genes were identified based on cells derived from the eight control mice using the FindAllMarkers function (logfc.threshold = 0.5, only.pos = TRUE. min.pct = 0.3, test.use = " wilcox"). Clusters were inspected and some very similar clusters were merged, leaving 11 distinct clusters, and cluster markers recalculated. Cluster sizes were quantified as a proportion of the total number of cells per mouse, and differences between control and infected (timepoint-matched) mice assessed with a two-tailed *t* test. Significantly enriched GO terms associated with cluster marker genes (Supplementary Data 3) were identified with the enrichGO function from the clusterProfiler package (v3.14.0).

To scrutinise the undifferentiated cells further, three undifferentiated clusters were isolated in silico. Feature counts were normalized and scaled, top 2000 variable features identified and PCA performed. The first 30 principal components were used for running UMAP for visualisation (min.dist = 0.05) and for neighbour finding, and clusters identified with the FindClusters function (resolution = 0.4). After merging of very similar clusters, five clusters remained and markers were recalculated as described above.

Libraries from eight control mice were further analysed with the monocle3 (v0.2.1) package[79] to place the cells in pseudotime. We applied a standard workflow with preprocess_cds, align_cds (correcting for batch), reduce_dimension (reduction_method = UMAP), cluster_cells and learn_graph to fit a trajectory graph. Two main partitions were identified, consisting of undifferentiated cells/enterocytes and goblet cells. Cells in the G2M phase were selected as the root of the trajectories for pseudotime ordering. We used partition-based graph abstraction (PAGA)[80], implemented in SCANPY[81] (v1.4.6), to draw an abstract connectivity graph. After standard pre-processing, regressing out sample batches, the PAGA graph was computed and visualized (threshold = 0.3).

**Reporting summary.** Further information on research design is available in the Nature Research Reporting Summary linked to this article.

**Data availability**

All the data needed to make the conclusions from this study are present in the main manuscript or supplementary materials. The transcriptomic datasets generated during and analysed the current study are available in the European Nucleotide Archive (ENA) repository (https://www.ebi.ac.uk/ena/browser/home), under the accession numbers ERP008531 (STDY 3371 - Mouse and *Trichuris muris* transcriptome time course, https://www.ebi.ac.uk/ena/browser/view/PRJEB7610?show=reads), ERP126662 (STDY 4023 - Investigating the transcriptome of early infective larvae stage of *Trichuris muris*,

https://www.ebi.ac.uk/ena/browser/view/PRJEB42759?show=reads) and ERP021944 (STDY 4672 - Investigating early host intestinal epithelial cells - whipworm interactions, https://www.ebi.ac.uk/ena/browser/view/PRJEB19877?show=reads). Databases used for the analysis are Ensembl (https://www.ensembl.org/index.html), the molecular signatures database (http://www.gsea-msigdb.org/gsea/msigdb/) and WormBase Parasite (https://parasite.wormbase.org/index.html). Source data are provided with this paper.

## Code availability

The R code used to analyse the bulk and single-cell RNA-seq data of this study is available in the Github repository https://github.com/fayerodgers/single_cell, https://doi.org/10.5281/zenodo.5984092[82].

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

## Acknowledgements

This work was supported by a National Centre for the Replacement, Refinement and Reduction of Animals in Research (UK) David Sainsbury Fellowship (grant NC/P001521/1, M.A.D-C.); the Medical Research Council (grant MR/R002800/1, D.J.T.); and the Wellcome Trust (grant numbers 206194, M.B.; Z10661/Z/18/Z, R.K.G.; and 088785/Z/09/Z, D.J.T. and R.K.G.). For the purpose of Open Access, the author has applied a CC BY public copyright licence to any Author Accepted Manuscript version arising from this submission.

## Author contributions

Conceptualization, M.A.D-C., R.K.G. and M.B; data curation, M.A.D-C., D.G., F.H.R., K.R., J.A.G., A.J.R., C.S., C.R. and T.S.; formal analysis, M.A.D-C., D.G., F.H.R., K.R., J.A.G., A.J.R., C.S. and C.R.; funding acquisition, M.A.D-C., D.J.T., R.K.G. and M.B.; investigation, M.A.D-C., D.G., C.C., K.R., J.A.G., A.J.B., H.M.B., M.E.L., A.S., P.S., N.R., C.T., C.M., C.B., C.S., C.R., J.G.M., C.M.C., T.S. and K.S.H.; methodology, M.A.D-C., D.G., F.H.R., K.R., J.A.G., A.J.B., H.M.B., A.J.R., A.S, P.S., N.R., C.S., C.R., T.S. and K.S.H.; project administration, M.A.D-C., N.H., M.S. and M.B.; resources, M.A.D-C., J.A.G., D.J.T., R.K.G. and M.B.; supervision, M.A.D-C., D.J.T., R.K.G. and M.B.; validation, M.A.D-C., D.G., K.R., J.A.G., C.S. and C.R.; visualization, M.A.D-C., D.G., F.H.R., K.R., J.A.G., A.J.R., C.S. and C.R.; writing of the original draft, M.A.D-C., F.H.R., K.R., J.A.G., H.M.B., A.J.R., D.J.T., R.K.G. and M.B.; writing–review and editing, M.A.D-C., F.H.R., K.R., J.A.G., D.J.T., R.K.G. and M.B.

## Competing interests

The authors declare no competing interests.
