## [Peer Review File · Nature Communications]

Defining the early stages of intestinal colonisation by whipwormsReviewers' Comments:

Reviewer #1:

Remarks to the Author:

In this study, the authors have investigated a previously mysterious step in *Trichuris* parasite infection. How does this parasite establish a partially intracellular and partially extracellular niche in the intestine? Technological advances, including the generation of caecaloids, as well as scRNA-seq has enabled the investigators to establish some interesting and novel findings from this unique host-parasite interaction. While previous microscopy studies have shown that the parasites burrow through several intestinal epithelial cells to form a syncytial tunnel, how this forms has not been well characterized. Surprisingly, the investigators found that the enterocytes and goblet cells that form this tunnel remain alive and interact with the larvae during infection, before triggering a type-1 IFN response with a signal from Isg15. This is surprising, because the general dogma is that alarmins such as IL-25, TSLP and IL-33 are dominant during helminth induced breaches of the epithelial barrier. Hence, overall, I found this to me a very interesting study that is a great starting point for many future studies and represents a significant advance to this field of research. A few points that could be addressed below could substantially improve the manuscript.

It is not entirely clear to me how the authors used TEM to identify and separate out the enterocytes present in the cecaloids from other associated epithelial cells present, for example transit amplifying cells, enteroendocrine cells and etc..

For some of the images, such as in Figure 2, including "negative control" uninfected/unaffected sections of either the same cecaloid or from a different uninfected cecaloid as a side-by-side image would be useful. Some form of quantification for goblet-cell hyperplasia/congregation around the worm in an uninfected/naïve cecaloid versus an infected one would provide more objective evidence to support the basic observations.

In the same vein, some form of quantification to support the observation for "loss of denseness of the mucin layer in the cecum following *Trichuris* infection" based on image analysis tools or other approach would provide greater confidence to these conclusions.

In Figure 4, the authors described a cell death induced migration in areas where the worms have transversed. Can this be reproduced in the caecaloid system? This would be important for validating the cecaloid system as a great model for in vitro *Trichuris* infection.

In the scRNA-seq data, it was surprising that Paneth cells was not represented. Would the authors care to comment on this? Are they depleted by *Trichuris* infection, or is it a coverage issue? It was also surprising that goblet cells were not substantially increased or altered based on the earlier conclusion of these cells being increased. These discrepancies may warrant further discussion.

An obvious next step would be to infected ISG15^{-/-} mice to validate that observation. This could be beyond the scope of this manuscript, but is an important future direction.

A few other minor issues:

Are the differences in the brightness and contrast of the toluidine staining in Figure 1b 3 h pi and 72 hr, a reflection of difference between the time points or a reflection of differences between staining protocols?

In page 9, line 238-245, the authors should refer to these as crypt-villi units rather than crypts alone especially as there are more distinct staining in the villi than in the crypts.

Reviewer #2:

Remarks to the Author:

In their manuscript „Defining the early stages of intestinal colonisation by whipworms“ the authors describe how first-stage larvae invade the host intestinal epithelium from mucus degradation to formation of syncytial tunnels. Interestingly, Duque-Correa et al. describe a new in vitro system “caecaloids” that can mimic this early stage of infection. Furthermore, the authors apply single-cell sequencing to explore the worm-epithelium interaction and report type-I IFN related epithelial response.

The manuscript is clear and well written. Most of the claims the authors make is supported by their data with some exceptions, where further clarification would be necessary (see specific points below). In particular, the caecaloid in vitro system to model early whipworm infection (probably one of the most interesting aspects of the manuscript) will need further characterization and application examples.

Specific points:

1) The authors claim throughout the manuscript that larvae specifically invade epithelial cells at the crypt base and predominantly enterocytes and goblet cells. This needs some closer investigation. The bottom of intestinal crypts are made up of Paneth cells (SI) or deep crypt secretory cells (colon) that serve as niches for stem cells that are wedged between them. Enterocytes and goblet cells start to appear above the stem cell zone. Hence, there is a disconnect between the claim that worms weave through enterocytes and goblet cells and the claim that it happens at the bottom of the crypt. Intestinal histology is inherently difficult since a cutting plane is unlikely to intersect the crypt at the ideal point to see the whole crypt villus structure, so the actual bottom of the crypt is easily cut off. To prove their claims the author need to stain their histological sections with actual markers of the stem cell zone to prove the localization of the worm. For this purpose, LYZ1 (if expressed by niche cells in caecum, otherwise REG4), LGR5 (only in situ hybridization, due to lack of good antibodies) or OLFM4 (for general stem cell/TA zone) would be suitable. Use of immunofluorescence (RNAscope for LGR5) would be preferable, due to better quality of the signal localization, but also immunohistochemistry would be acceptable. Additionally, these data should be further complement by goblet (MUC2) and colonocyte (e.g. carbonic anhydrase I) stainings on the histological sections to clearly show invasion of these cells.

2) The authors claim that caecaloids “generated self-organizing structures that resembled crypts present in the caecum”. Since the authors describe a new in vitro system here, this has to be clearly demonstrated. In fact, I am surprised to hear that caecaloid did form crypt structures, since organoids on transwells usually do not do this, but rather build a 2D layer with (at best) limited zonation. None of the provided data shows this crypt formation. The same histological analysis mentioned in point 1, should be applied to vertical sections (not the provided horizontal ones) of the caecaloid system to clearly show crypt structure, stem cell location and differentiated cells at low (for general overview) and high magnification.

3) Since caecaloids make the invasion process accessible, the author should consider to perform live-imaging of the invasion, which would strongly corroborate their description of worm invasion.

4) Generally, the use of the caecaloid system should be further expanded to demonstrate the potential of the system. For example the authors should quantify whether addition of certain drugs to the system can prevent initial worm invasion (e.g. Albendazole or serine endopeptidase inhibitors) to show potential future applications for targeted drug development.

5) The authors should describe how long the whipworm can be followed in the caecaloid system. Can molting be observed? If stability of the caecaloid is the limitation of the system, can worms be isolated and reseeded onto a fresh transwell to bring them eventually to maturity? These questions will be

important to answer to properly judge the value of the in vitro system for future research.

Reviewer #3:

Remarks to the Author:

The work presented by Duque-Correa et al. is a nice description of early stage L3 *T. muris* in/making tunnel syncytia in the colon. The images presented are nice and convincing of the helminth in a host cell tunnel. This said, I am not convinced that the work makes a substantial contribution to the field. The observations as presented look great but there is no indication of how representative the images are, that is, what is the reproducibility. The use of cecealoids demonstrates similar phenomena in vitro but does not provide mechanistic insight. The authors suggest the worm preferentially burrows into enterocytes and goblet cells in the crypts (again no numeric data), but what of Paneth cells or stem cells? Are these cells purposefully avoided, and if so what cue might the worm be responding too? The authors indicate that the worm can degrade mucus, and while this is an intuitive finding it is nevertheless good to have data supporting this. However, again the investigators stop short and provide no experiments relating to how the mucus degradation is achieved. They make interesting comments e.g. "the infected cells have lots of mitochondria." Is this more than uninfected cells, presumably so since they comment but there is no quantification. What are the functional consequences of this finding?

The single cell RNAseq may in fact be the first demonstration in these cells/models but it seems like adding a specialized technique to a descriptive study and providing more descriptive data. There is no confirmation of the findings by another technique and therefore the discussion is speculative. The abstract states, "characterised by the expression of Isg15, instigating the host immune response". This seems like a stretch. What is the evidence for this apart from the correlation of increased Isg15 in tissue from infected mice?

Clearly, the authors have put considerable effort into the tracking and imaging of *T. muris* in situ and despite the impressive images, the manuscript as a whole does not provide substantial insight to the host-parasite interaction. I have concerns around reproduce-ability (i.e. n values), absence of mechanistic studies, and largely speculative conclusions based on descriptive data. The authors are to be congratulated for the technical aspects of this work and the images they provide, yet to me, the material is perhaps better suited to a parasitology journal.

We thank the Reviewers from their extensive and insightful comments on our manuscript (**NCOMMS-21-06904**). A particular area of focus for us has been on strengthening the experimental validation of the caecaloids. Some of our options have been limited due to restricted working conditions but we believe we have substantially improved the manuscript. Detailed responses to each point raised are listed below but we would like to particularly draw attention to our new data, extensively cross validating marker genes in caecaloids and caecum.

Reviewer #1 (Remarks to the Author):

In this study, the authors have investigated a previously mysterious step in *Trichuris* parasite infection. How does this parasite establish a partially intracellular and partially extracellular niche in the intestine? Technological advances, including the generation of caecaloids, as well as scRNA-seq has enabled the investigators to establish some interesting and novel findings from this unique host-parasite interaction. While previous microscopy studies have shown that the parasites burrow through several intestinal epithelial cells to form a syncytial tunnel, how this forms has not been well characterized. Surprisingly, the investigators found that the enterocytes and goblet cells that form this tunnel remain alive and interact with the larvae during infection, before triggering a type-1 IFN response with a signal from Isg15. This is surprising, because the general dogma is that alarmins such as IL-25, TSLP and IL-33 are dominant during helminth induced breaches of the epithelial barrier. Hence, overall, I found this to me a very interesting study that is a great starting point for many future studies and represents a significant advance to this field of research. A few points that could be addressed below could substantially improve the manuscript.

We thank the reviewer for the appreciation of the novelty of our caecaloid model for *Trichuris muris* and our findings on the early host intestinal epithelial-whipworm interactions that allow the parasite to establish in its niche.

- 1) It is not entirely clear to me how the authors used TEM to identify and separate out the enterocytes present in the caecaloids from other associated epithelial cells present, for example transit amplifying cells, enteroendocrine cells and etc.

We agree that by using TEM it is difficult to identify and distinguish stem, progenitors and TA cells from enterocytes, and deep secretory cells (DSCs) from goblet cells (both contain mucous secretory granules). In contrast, enteroendocrine cells are a very distinctive cellular population identified by presence of punctate black secretory granules (current *Supplementary Fig. 4h*).

Our single-cell RNAseq dataset allowed us to characterise the different cell types present in the caecum for the first time (current *Fig. 8* and *Supplementary Figs. 10* and *11*). Using these data, we have now selected marker genes for each cellular population and performed combined *in situ* mRNA hybridization (ISH) with immunofluorescence (IF) by chain reaction (HCR) to visualise each cell type in both the caecum and caecaloids (new *Supplementary Figs. 2* and *5*).

The new marker data, combined with imaging presented in *Supplementary Fig. 4*, and our previous results¹, have greatly expanded our characterisation of the cell populations, confirming that the cellular composition and organisation are common to the caecaloids and caecum. Moreover, the improved resolution with new markers has allowed us to better identify the cellular populations associated with whipworm L1 larvae (new *Figs. 1* and *3*), namely, mitotically active cells comprising stem, TA, DSCs and enterocyte progenitors. Supported by these findings, we have now re-interpreted our TEM/serial block face SEM results and classified the cells with mucus secretory granules as DSCs (current *Fig. 2*,

Supplementary Fig. 1). However, we cannot distinguish among stem, TA and progenitor cells. To reflect these changes, we have now modified the main text in lines 39, 91, 100, 105-122, 126, 129, 138-139, 151-176, 290, 325-328 and 357-358, and *Figs. 2 and 4 and Supplementary Fig. 1*.

- 2) For some of the images, such as in Figure 2, including “negative control” uninfected/unaffected sections of either the same cecaloid or from a different uninfected cecaloid as a side-by-side image would be useful. Some form of quantification for goblet-cell hyperplasia/congregation around the worm in an uninfected/naïve cecaloid versus an infected one would provide more objective evidence to support the basic observations.

In addition to confocal IF microscopy and TEM images from uninfected caecaloids in *Supplementary Fig. 4*, where we have characterised the different cellular populations present in this model, we now include characterisation using ISH coupled with IF by HCR (new *Supplementary Fig. 5*).

There has been a misunderstanding regarding our observations on whipworm L1 larvae infecting goblet cells. While we showed the larvae in the cytoplasm of cells with secretory granules including both DSCs and goblet cells (current *Figs. 3 and 4*), we are not suggesting that there is goblet-cell hyperplasia/congregation around the worm. Goblet cell hyperplasia is a phenomenon occurring later during mouse infections (3 weeks p.i.), when cytokine-dependent resistance mechanisms develop¹. Unfortunately, we have not modelled this phenomenon in caecaloids yet and in our current study we focus on observations during the first three days of infection.

- 3) In the same vein, some form of quantification to support the observation for “loss of denseness of the mucin layer in the cecum following *Trichuris* infection” based on image analysis tools or other approach would provide greater confidence to these conclusions.

Following the suggestion of the reviewer we have measured the density of toluidine blue staining, which is a metachromatic dye especially selected for its high affinity to acidic components, particularly mucins, by quantifying the %CMYK recorded using the counting tool in Adobe Photoshop. We have counted five data points from each of three areas: 1) mucus layer overlaying uninfected IECs, 2) cleared mucus above IECs infected with whipworm larvae, and 3) mucus-free background (above and away from the IECs section) for each of 5 worms. We observed a significant statistical difference in the %CMYK when comparing the mucus above regions infected with whipworm larvae with the mucus overlaying uninfected regions. This data has been included as new *Fig. 5d*, and clearly supports our observations on the loss of denseness of the mucus layer overlaying infected cells.

- 4) In Figure 4, the authors described a cell death induced migration in areas where the worms have transversed. Can this be reproduced in the caecaloid system? This would be important for validating the cecaloid system as a great model for in vitro *Trichuris* infection.

We are pleased the reviewer highlighted visualisation of cell death-induced migration as being an important use of the caecaloid model. The images presented in *Fig 4a* (current *Fig 6a*) are in fact from caecaloids. To clarify this, we have modified the sentence in line 218 as follows:

“In contrast, at early stages of infection of caecaloids, we found that while cells left behind in the tunnel were dead, the IECs actively infected by the worm were in fact alive (Fig. 6a).”

It is not possible to visualise the tunnels burrowed by whipworm L1 larvae *in vivo* as the larvae are completely embedded in the intestinal epithelium and the parasite is currently genetically intractable². However, the open conformation of our caecaloid-whipworm model system allowed us to study early whipworm interactions with the epithelial cells to a level previously unattainable, revealing the intricate path of the tunnels formed by L1 larvae. Supporting the confocal microscopy data on cell death, TEM images showed cell liquefaction, and pyknotic nuclei indicating early apoptosis in the caecaloids (current Figs. 6c and d), which are also observed *in vivo* (current Figs. 2d and e).

- 5) In the scRNA-seq data, it was surprising that Paneth cells was not represented. Would the authors care to comment on this? Are they depleted by *Trichuris* infection, or is it a coverage issue?

We are confident that this is not a coverage issue because our single-cell data readily detected populations with very low proportions, such as tuft (0.16%) and enteroendocrine cells (0.18%) (Fig 8).

The caecal epithelium is different from that of the small intestine and an absence of Paneth cells has been reported³⁻⁵. We have previously characterised this difference by staining Lysozyme 1 (LYZ1), a marker of Paneth cells, in both small intestinal and caecal epithelia as well as enteroids and caecaloids³ (See *Supplementary Fig 1* from the referenced manuscript below). Strikingly, organoids closely recapitulate the cellular composition of the tissue of origin.

Supplementary Figure 1 (from Duque-Correa et al 2020). Lysozyme staining as a marker of Paneth cells in small intestine and caecum tissue and organoids. Images of confocal IF microscopy with an antibody staining Lysozyme-expressing Paneth cells in small intestinal tissue and organoids (A), which are not frequently found in caecal tissue and organoids (B). DAPI stains nuclei and DiI the cell membranes. Scale bars 100 μ m.

Moreover, in our single-cell RNAseq dataset we do not detect any expression of *Lyz1* in caecal IECs:

To clarify the lack of Paneth cells in the caecal epithelium, we have now included the following sentence in the manuscript (lines 58-60):

“The epithelial composition of the caecum has not been studied at the single-cell level. However, histological studies have shown that it lacks Paneth cells and has numerous goblet cells, although fewer than colonic epithelium^{6,7}.”

- 6) It was also surprising that goblet cells were not substantially increased or altered based on the earlier conclusion of these cells being increased. These discrepancies may warrant further discussion.

Following up on our previous clarification (point 2), while we observed whipworm L1 larvae infecting cells with mucous secretory granules (now identified as DSCs), we did not say these cells are increased upon infection. However, it is interesting we did not find a particular response of DSCs or other mitotically active cells to infection, but identified the expansion of a population of enterocytes expressing *Isg15*, which are not infected by the larvae (Figs. 8 and 9). This suggests that bystander cells respond to cell damage caused by the larvae tunnelling through IECs at the crypt base or to the larval antigens, and this has been discussed in the manuscript (lines 355-379).

- 7) An obvious next step would be to infected ISG15^{-/-} mice to validate that observation. This could be beyond the scope of this manuscript, but is an important future direction.

We agree with the reviewer this is an important future direction, clearly the next thing to do and we are currently working on it. However, we consider such studies to be beyond the scope for the current manuscript, which already includes a vast amount of data identifying novel early interactions between IECs and whipworms, visualising the early syncytial tunnels, characterising for the first time the caecal epithelium at single-cell level, and describing the first *in vitro* model for whipworms and organoid model for live helminths.

A few other minor issues:

- 8) Are the differences in the brightness and contrast of the toluidine staining in Figure 1b 3 h pi and 72 hr, a reflection of difference between the time points or a reflection of differences between staining protocols?

The differences in the brightness and contrast of the toluidine staining are not a reflection of the difference between the time points. They are the same stain but different experiments so they can appear variable because we find the worms by looking retrospectively through literally several hundreds, sometimes thousands of sections that have been cut serially and stained *en masse*. Staining reproducibility is difficult depending on the stain batch, user and sample.

- 9) In page 9, line 238-245, the authors should refer to these as crypt-villi units rather than crypts alone especially as there are more distinct staining in the villi than in the crypts.

Different from the small intestinal and similar to the colonic epithelium, the caecal epithelium lacks villi and consists of crypts of Lieberkühn with only short regions of flat surface epithelium^{3,4}. We therefore refer to the crypt bottom/base, the crypt top and the intercrypt table, but not villi.

In order to provide the reader with a clearer description of the unique features of the caecal epithelium, we have now added the following sentences in the introduction (lines 55-62):

“The caecal epithelium is a distinctive tissue; it lacks the villi present in the small intestinal mucosa but, similar to the colonic epithelium, it is composed of a flat epithelial surface (known as the intercrypt table) within which crypts of Lieberkühn are embedded⁶⁻⁸. The epithelial composition of the caecum has not been studied at the single-cell level. However, histological studies have shown that it lacks Paneth cells and has numerous goblet cells, although fewer than colonic epithelium^{6,7}. The resulting differences in the mucus layers overlying the caecal mucosa and in the microbiota harboured within the caecum^{7,9,10}, have created a distinct niche in which whipworms have evolved to invade and persist.”

Reviewer #2:

In their manuscript “Defining the early stages of intestinal colonisation by whipworms” the authors describe how first-stage larvae invade the host intestinal epithelium from mucus degradation to formation of syncytial tunnels. Interestingly, Duque-Correa et al. describe a new in vitro system “caecaloids” that can mimic this early stage of infection. Furthermore, the authors apply single-cell sequencing to explore the worm-epithelium interaction and report type-I IFN related epithelial response.

The manuscript is clear and well written. Most of the claims the authors make is supported by their data with some exceptions, where further clarification would be necessary (see specific points below). In particular, the caecaloid in vitro system to model early whipworm infection (probably one of the most interesting aspects of the manuscript) will need further characterization and application examples.

Specific points:

1) The authors claim throughout the manuscript that larvae specifically invade epithelial cells at the crypt base and predominantly enterocytes and goblet cells. This needs some closer investigation. The bottom of intestinal crypts are made up of Paneth cells (SI) or deep crypt secretory cells (colon) that serve as niches for stem cells that are wedged between them. Enterocytes and goblet cells start to appear above the stem cell zone. Hence, there is a disconnect between the claim that worms weave through enterocytes and goblet cells and the claim that it happens at the bottom of the crypt.

Intestinal histology is inherently difficult since a cutting plane is unlikely to intersect the crypt at the ideal point to see the whole crypt villus structure, so the actual bottom of the crypt is easily cut off. To prove their claims the author need to stain their histological sections with actual markers of the stem cell zone to prove the localization of the worm. For this purpose, LYZ1 (if expressed by niche cells in caecum, otherwise REG4), LGR5 (only in situ hybridization, due to lack of good antibodies) or OLFM4 (for general stem cell/TA zone) would be suitable. Use of immunofluorescence (RNAscope for LGR5) would be preferable, due to better quality of the signal localization, but also immunohistochemistry would be acceptable. Additionally, these data should be further complemented by goblet (MUC2) and colonocyte (e.g. carbonic anhydrase I) stainings on the histological sections to clearly show invasion of these cells.

We agree with the reviewer there is a disconnection between the claims of the whipworm L1 larvae infecting the crypt bottom and enterocytes and goblet cells. Following the suggestions of the reviewer and to identify the stem cell zone, we performed ISH HCR for *Lgr5*. However, while *Lgr5* is detectable by ISH HCR in the small intestine (see image below), it is not in the caecum.

Moreover, according to our single-cell RNAseq data, caecal IECs do not express *Olfm4* or *Lyz1*:

UMAP plots from single-cell RNA-seq analysis showing normalized expression level of *Olfm4* and *Lyz1* in IECs from the caecum of uninfected mice (n=8).

In addition, we have previously shown that Paneth cells (LYZ1 positive) are not present in the caecum or caelocaloids^{3,4} (see response to Reviewer 1, point 5).

Therefore, we next mined our single-cell RNA-seq data to identify markers for the different cellular populations of the caecum. Expression data for the chosen markers are shown in detail below but also summarised in the manuscript (lines 105-116, current *Fig. 8*, *Supplementary Figs. 2, 10-12, Supplementary File 3*):

Violin plots showing expression (y-axis) of selected marker genes of cellular populations from the caecum of uninfected mice (n=8).

Those populations are:

1. **Stem cells**, identified by the expression of *Ascl2*, which is a marker of stem cells in the colon^{6,7}. However, in the caecum this gene is also expressed by DSCs, which we observed by ISH HCR (new *Supplementary Fig. 2*).
2. **Transamplifying (TA)/dividing cells**, characterised by the expression of *Mki67*, which we detected by IF HCR using an antibody against Ki67, but lacking expression of any other of the selected cell type markers. ISH HCR probes for other marker genes such as *Tob2a*, *Birc5*, *Stmn1* and *Ube2c* did not work in our hands.
3. **DSCs**, identified by the expression of *Ascl2*, *Reg4*, *Muc2* and *Mki67*.

4. **Enterocyte progenitors**, characterised by the expression of *Mki67* and *Car1*.
5. **Enterocytes**, expressing *Car1* and *Krt20*. Note that, while *Car1* is highly expressed by enterocyte progenitors and enterocytes, it is also expressed by all the other cell types.
6. **Goblet cells**, identified by the expression of *Muc2*, and binding of the lectins *Ulex europaeus* agglutinin (UEA) and *Sambucus nigra* (SNA). Note that according to our single-cell-RNAseq data, *Reg4* is both expressed by DSCs and goblet cells in the caecum. However, we only observed *Reg4* expression in DSCs using ISH HCR (new *Supplementary Fig. 2*).

Using ISH and IF by HCR we visualised each cell type on histological sections of the caecum, which has allowed us to clearly identify the bottom of the crypt and the cellular populations in the crypt's "dividing" and "differentiated" zones (new *Supplementary Fig. 2*). Moreover, by co-staining with *p43*, the single most abundant protein secreted/excreted by the parasite⁸, we have found that *T. muris* L1 larvae are associated predominantly with mitotically active cell types in the base of mouse caecal crypts including stem, DSC and TA cells as well as enterocyte progenitors (new *Fig. 1*).

Taking into account these new data, we have re-interpreted our TEM and serial block face SEM results and have modified the figures accordingly. Cells previously identified as goblet cells and enterocytes, are now identified as DSCs and other dividing cells, respectively (current *Fig. 2* and *Supplementary Fig. 1*).

2) The authors claim that caecaloids "generated self-organizing structures that resembled crypts present in the caecum". Since the authors describe a new *in vitro* system here, this has to be clearly demonstrated. In fact, I am surprised to hear that caecaloid did form crypt structures, since organoids on transwells usually do not do this, but rather build a 2D layer with (at best) limited zonation.

None of the provided data shows this crypt formation. The same histological analysis mentioned in point 1, should be applied to vertical sections (not the provided horizontal ones) of the caecaloid system to clearly show crypt structure, stem cell location and differentiated cells at low (for general overview) and high magnification.

The reviewer is correct, we have not reproduced three-dimensional crypts in our model. Our choice of the word "resembled" is open to interpretation and falsely created the impression of near-identical 3D conformation. We have therefore toned down the language used. However, the caecaloids grown in transwells do recapitulate many aspects of the crypt's epithelial cell-type composition and organisation. Echoing the crypt base, we see centres of dividing/proliferative (*Ki67*⁺) cells that are surrounded by large areas of non-proliferative, differentiated cells (current *Supplementary Fig. 4*).

Following the suggestion of the reviewer, we have now used ISH/IF by HCR to image all the cellular populations we have stained in caecum by doing whole mount staining and microscopy of caecaloids (new *Supplementary Fig 5*). In these new experiments, we observed that within the centres of proliferative cells (*Ki67*⁺), stem cells (*Ascl2*⁺) reside adjacent to DSCs (*Muc2*⁺) cells, recapitulating their spatial arrangement *in vivo* (new *Supplementary Figs. 5a-c*). Goblet cells (*Muc2*⁺, UEA⁺) and enterocytes (*Car1*⁺) surround these centres (new *Supplementary Figs 5d-g*).

Whipworm L1 larvae were found infecting cells in proliferating centres and adjacent areas in the caecaloids (new *Fig. 3*), effectively reproducing *in vivo* infection and the interaction of the

parasite with the IECs in the dividing zone of the caecal crypts. To reflect these new findings and add quantification of our observations, the main text has been updated, lines 151-169.

3) Since caecaloids make the invasion process accessible, the author should consider to perform live-imaging of the invasion, which would strongly corroborate their description of worm invasion.

While we would love to perform live-imaging of whipworm invasion, that experiment is currently well beyond the bounds of technical feasibility. First, no transgenesis is currently possible in whipworms; therefore, we cannot reliably stain live larvae and follow them during invasion. Second, we have not yet developed a vessel to support direct live-imaging of the caecaloids grown in transwells, and their current configuration will make it impossible for the microscope objectives to reach inside the transwell membrane where the cells are.

Nevertheless, our confocal and TEM data clearly show the whipworm L1 larvae have invaded the cells, we see the worms are in direct contact with the IECs cytoplasm (new Figs. 3-4). Thus, we consider that while live imaging would provide nice videos on how the invasion process occurs, scientifically, it wouldn't in fact provide additional knowledge beyond what we are currently showing.

4) Generally, the use of the caecaloid system should be further expanded to demonstrate the potential of the system. For example the authors should quantify whether addition of certain drugs to the system can prevent initial worm invasion (e.g. Albendazole or serine endopeptidase inhibitors) to show potential future applications for targeted drug development.

It would be an extraordinarily large undertaking to incorporate drug-inhibition assays into this work (likely measured in years) and we believe it is beyond the scope of this study. This is the first example of a metazoan parasite interacting, infecting and modifying an organoid, and has already allowed us to better understand the early interactions of whipworms with the intestinal epithelia in a level unattainable *in vivo*.

Adding albendazole to the larvae prior to invasion to evaluate if it can prevent invasion will not be informative as others have shown the low activity of various benzimidazoles by direct exposure of L1 larvae *in vitro*⁹. Moreover, serine peptidase inhibitors are of broad action and can affect other proteases on the larvae, not only those potentially involved in invasion. Ideally, one would generate knock-out larvae for the serine endopeptidase genes identified in our study; however, transgenesis is still not available for whipworms. We have discussed these studies as future directions in lines 328-331.

5) The authors should describe how long the whipworm can be followed in the caecaloid system. Can moulting be observed? If stability of the caecaloid is the limitation of the system, can worms be isolated and reseeded onto a fresh transwell to bring them eventually to maturity? These questions will be important to answer to properly judge the value of the *in vitro* system for future research.

So far, we have followed whipworm infection in the caecaloids for 13 days. *In vivo* the first moult occurs between days 9 and 11 post infection (p.i)¹ and *in vitro* we have observed L2 larvae at days 11 and 13 p.i. in two independent experiments using two different caecaloid lines. While in the majority of the transwells we only found L2 larvae, in some of them there was still a combination of L1 and L2 whipworms (see image below). L2 larvae are considerably bigger than L1 larvae but remain completely intracellular.

Representative confocal IF images of caecaloids infected with *T. muris* for 11 days. Complete z-stack **(a)** and selected and cropped volume **(b)** showing L2 (upper worm) and L1 (lower worm) larvae infecting IECs within or adjacent to Ki-67⁺ (red) dividing centres. In green, the lectins UEA and SNA bind mucins in goblet cells; in blue DAPI stains nuclei of IECs and larvae; and in white, phalloidin binds to F-actin. Scale bars 20 μ m.

However, the application of the whipworm-caecaloid system to evaluate moulting is not the main aim of the current manuscript. Thus, we have decided not to publish these results in this paper but to include them on a follow-up publication of current work.

Reviewer #3:

The work presented by Duque-Correa et al. is a nice description of early stage L3 *T. muris* in/making tunnel syncytia in the colon. The images presented are nice and convincing of the helminth in a host cell tunnel. This said, I am not convinced that the work makes a substantial contribution to the field.

We are disappointed that the reviewer does not recognise how our manuscript substantially contributes to helminthology and, specifically, Trichuriasis. The caecaloid model is the very first *in vitro* model for whipworms and organoid system for helminths. Moreover, we are the first to show and characterise the early tunnels generated by whipworm L1 larvae both in the caecum of mice and in the caecaloids.

- 1) The observations as presented look great but there is no indication of how representative the images are, that is, what is the reproducibility.

Quantification of *in vivo* experiments is a huge issue because of the constraints to locate intracellular whipworm L1 larvae, which infect IECs at the bottom of the crypts that account for less than 1% of the caecal epithelium, and the lack of genetic tools to generate fluorescent larvae. Nevertheless, our efforts have allowed us to find 46 *T. muris* L1 larvae during the first 72 h of infection across three independent experiments with three mice per time point each, which were evaluated by TEM/serial block face SEM (15 worms) and IF/ISH HCR (31 worms). In addition, we have included in the manuscript quantification of our observations for key points including serial block face SEM (lines 137-139, *Supplementary*

Fig. 1), desmosome separation (lines 242-246, current *Fig. 7* and *Supplementary Fig. 8*) and now for association with cells at the bottom at the crypts (lines 120-122, new *Fig. 1*).

One of the big advantages of the caecaloids system is that there are dozens, if not hundreds, of L1 larvae infecting a transwell alone. Moreover, our experiments have been done in triplicate across more than ten independent replicas using three caecaloid lines derived from three C57BL/6 mice. In our latest ISH HCR experiment on caecaloids, we quantified the association of L1 larvae with different cell types and have now included these figures on lines 161-169.

To indicate how representative our images are, we have now included these metadata in the figure legends of *Figs. 1-7*.

- 2) The use of caecaloids demonstrates similar phenomena *in vitro* but does not provide mechanistic insight.

While strictly speaking our findings are not mechanistic, they are still novel biological and immunological insights. This is the first-time organoids sustain an intracellular helminth parasite and characterising the model in terms of the reproducibility of the *in vivo* infection was therefore a crucial milestone. Our data has started to build a landscape on how the parasite invades the host intestinal epithelia and now we are in the position to move forward to test potential mechanisms of invasion and colonisation with the caveat that transgenesis of whipworms is still not possible.

- 3) The authors suggest the worm preferentially burrows into enterocytes and goblet cells in the crypts (again no numeric data), but what of Paneth cells or stem cells? Are these cells purposefully avoided, and if so what cue might the worm be responding too?

Following the suggestions from Reviewer 1 (point 1) and 2 (point 1), we have generated new imaging data using ISH/IF HCR to detect the different cellular populations of the caecal epithelium and better identify the cells associated with whipworm L1 larvae (New *Figs. 1, 3* and *4* and *Supplementary Figs 2, 4* and *5*). These new experiments allowed us to re-interpret our TEM data and suggest that the worms preferentially infect mitotically active cells at the bottom of the crypt, including stem, TA, DSC cells and enterocyte progenitors. Accordingly, we have now modified the description of our results in 39, 91, 100, 105-122, 126, 129, 138-139, 151-176, 290, 325-328 and 357-358.

These results lead us to hypothesise that the cue the worm may respond to specific cues from proliferative cells in order to infect them. This has been discussed in lines 325-328.

- 4) The authors indicate that the worm can degrade mucus, and while this is an intuitive finding it is nevertheless good to have data supporting this. However, again the investigators stop short and provide no experiments relating to how the mucus degradation is achieved.

In our manuscript we present data supporting mucus degradation during the invasion process. First, we detected whipworm L1 larvae expressing serine proteases at 3 and 24 h p.i. of mice (current *Supplementary Fig. 6a*); second, we showed mucin depolymerization of purified glycosylated MUC2 by L1 larvae *in vitro* (current *Fig. 5a*); and finally, we detected degradation of mucus of infected caecaloids (current *Figs. 5b-d, Supplementary Fig. 7*).

The primary purpose of our study was to generate and extensively validate an *in vitro* system to study early whipworm-IECs interactions that cannot be investigated *in vivo*, including mucus degradation. While we are currently following up our results to find a mechanism on how the mucus degradation is achieved, with the caveat that whipworm transgenesis is not available, we consider those experiments to be out of the scope of the current manuscript.

- 5) They make interesting comments e.g. “the infected cells have lots of mitochondria.” Is this more than uninfected cells, presumably so since they comment but there is no quantification. What are the functional consequences of this finding?

Although the reviewer is not referring to the core findings of the paper, we agree that our comment was rather anecdotal in nature. We have, therefore, now quantified mitochondria on our TEM images from *T. muris*-infected and uninfected cells in the caecum and caecaloids. No statistically significant differences were found.

We also did not find any significant difference in the size of mitochondria of infected cells compared to those of uninfected. Indeed, mitochondria size varies quite a lot from cell to cell. Thus, we cannot say that mitochondria are either more numerous or larger, in infected cells compared to uninfected.

Next, we measured the average thickness of the cytoplasm of uninfected cells and those infected with the whipworm L1 larvae. We found the cytoplasm of infected cells is thinner, which perhaps explains the appearance of concentrated mitochondria.

Considering these measurements, we have decided to remove the observations about mitochondria from the manuscript.

- 6) The single cell RNAseq may in fact be the first demonstration in these cells/models but it seems like adding a specialized technique to a descriptive study and providing more descriptive data. There is no confirmation of the findings by another technique and therefore the discussion is speculative.

We disagree with the reviewer. Our single-cell data has been validated using ISH HRC (current Figs. 8 and 9, novel Fig. 1 and Supplementary Fig. 2).

- 7) The abstract states, “, instigating the host immune response”. This seems like a stretch. What is the evidence for this apart from the correlation of increased Isg15 in tissue from infected mice?

We agree with the reviewer and have modified the text accordingly inserting the word 'potentially' in line 41.

Clearly, the authors have put considerable effort into the tracking and imaging of *T. muris* in situ and despite the impressive images, the manuscript as a whole does not provide substantial insight to the host-parasite interaction. I have concerns around reproduce-ability (i.e. n values), absence of mechanistic studies, and largely speculative conclusions based on descriptive data. The authors are to be congratulated for the technical aspects of this work and the images they provide, yet to me, the material is perhaps better suited to a parasitology journal.

The only available data on the early stages of whipworm infection are from light microscopy studies dating back 40-50 years, which have shown *T. muris* L1 larvae infecting cells at the base of the crypts of Lieberkühn in the first hours of an infection, but have not defined the host cells or identified any particular interactions^{2,10-12}. Our manuscript has provided not only a very detailed microscopic study of the host intestinal epithelia-whipworm interactions *in vivo* (reorganisation of cytoskeleton, apoptosis and liquefaction of cells, desmosome separation), with quantification for key points, but also novel transcriptomic studies showing for the first time the responses of IECs (e.g. expansion of Isg15 enterocyte population) and whipworms (e.g. distinct protease repertoires) during early infection. Moreover, we have developed the first *in vitro* model for whipworm infection, which is also the very first-time organoids have been used to study live helminths. Together these approaches have provided new insights on how the parasite invades and forms the syncytial tunnels inside live IECs.

We hope we have now addressed the reviewer's concerns about reproducibility (see response to point 1). We have been clear in sections of the manuscript where we have speculated; however, most of the mechanistic studies suggested by the reviewer would require technologies that do not yet exist or an amount of laboratory work out of the scope of the current manuscript.

References

- 1 Klementowicz, J. E., Travis, M. A. & Grecis, R. K. Trichuris muris: a model of gastrointestinal parasite infection. *Seminars in immunopathology* **34**, 815-828, doi:10.1007/s00281-012-0348-2 (2012).
- 2 O'Sullivan, J. D. B., Cruickshank, S. M., Withers, P. J. & Else, K. J. Morphological variability in the mucosal attachment site of Trichuris muris revealed by X-ray microcomputed tomography. *Int J Parasitol* **51**, 797-807, doi:10.1016/j.ijpara.2021.04.006 (2021).
- 3 Duque-Correa, M. A. *et al.* Development of caecaloids to study host-pathogen interactions: new insights into immunoregulatory functions of Trichuris muris extracellular vesicles in the caecum. *Int J Parasitol* **50**, 707-718, doi:10.1016/j.ijpara.2020.06.001 (2020).
- 4 Mowat, A. M. & Agace, W. W. Regional specialization within the intestinal immune system. *Nature reviews. Immunology* **14**, 667-685, doi:10.1038/nri3738 (2014).
- 5 Nguyen, T. L., Vieira-Silva, S., Liston, A. & Raes, J. How informative is the mouse for human gut microbiota research? *Dis Model Mech* **8**, 1-16, doi:10.1242/dmm.017400 (2015).
- 6 Parikh, K. *et al.* Colonic epithelial cell diversity in health and inflammatory bowel disease. *Nature* **567**, 49-55, doi:10.1038/s41586-019-0992-y (2019).
- 7 Murata, K. *et al.* Ascl2-Dependent Cell Dedifferentiation Drives Regeneration of Ablated Intestinal Stem Cells. *Cell Stem Cell* **26**, 377-390 e376, doi:10.1016/j.stem.2019.12.011 (2020).
- 8 Bancroft, A. J. *et al.* The major secreted protein of the whipworm parasite tethers to matrix and inhibits interleukin-13 function. *Nat Commun* **10**, 2344, doi:10.1038/s41467-019-09996-z (2019).
- 9 Keiser, J. & Haberli, C. Evaluation of Commercially Available Anthelmintics in Laboratory Models of Human Intestinal Nematode Infections. *ACS Infect Dis* **7**, 1177-1185, doi:10.1021/acsinfectdis.0c00719 (2021).
- 10 Panesar, T. S. The early phase of tissue invasion by Trichuris muris (nematoda: Trichuroidea). *Z Parasitenkd* **66**, 163-166, doi:10.1007/bf00925723 (1981).
- 11 Panesar, T. S. & Croll, N. A. The location of parasites within their hosts: site selection by Trichuris muris in the laboratory mouse. *Int J Parasitol* **10**, 261-273, doi:10.1016/0020-7519(80)90006-5 (1980).
- 12 Wakelin, D. The development of the early larval stages of Trichuris muris in the albino laboratory mouse. *J Helminthol* **43**, 427-436, doi:10.1017/s0022149x00004995 (1969).

Reviewers' Comments:

Reviewer #1:

Remarks to the Author:

The study investigators have carefully and extensively addressed all the comments that were raised by myself (as well as the other reviewers). As such, the manuscript is substantially improved and the work is both interesting, novel and a substantial advancement to the field.

Reviewer #2:

Remarks to the Author:

The authors addressed most points I raised during the revision. There are two minor points (see below) that would still need attention. Once these have been taken care of, I support publication.

Minor points:

1) the authors state that caecaloids "recapitulate caecal epithelia cell-type composition and crypt organization" (line 156). Given that the transwell model is essentially 2-dimensional, this is still misleading and should be corrected to not give wrong impressions about the presence of actual crypts.

2) The experiment testing for association between L1 larvae and proliferating centres lacks important information on the actual areas covered by proliferating and differentiated cells. This information is necessary to correctly interpret the numbers the authors provide. If the area of Ki67 positive cells is just 5% of the total area, this finding is highly specific. If 90% of the transwell-area is composed of Ki67 positive cells, even an association of 27/30 would merely indicate completely random distribution. According to the one overview picture the authors provide, the Ki67 positive area appears to be smaller compared to the differentiated one. Nevertheless, the authors should do proper statistics here that take the actual area of each region into account and tests for significance.

Reviewer #3:

Remarks to the Author:

Authors have responded satisfactorily to the prior comments.

Hereby we address **Reviewer #2** minor comments on our manuscript (**NCOMMS-21-06904**).

The authors addressed most points I raised during the revision. There are two minor points (see below) that would still need attention. Once these have been taken care of, I support publication.

Minor points:

1) the authors state that caecaloids “recapitulate caecal epithelia cell-type composition and crypt organization” (line 156). Given that the transwell model is essentially 2-dimensional, this is still misleading and should be corrected to not give wrong impressions about the presence of actual crypts.

Following the reviewer suggestion and taking as a guide manuscripts by Thorne, *et al.*¹ and Wang, *et al.*², we have modified the sentence in current lines 151-153:

Caecaloids cultured in an open conformation using transwells generated two-dimensional self-organizing structures that recapitulate caecal epithelia cell-type composition and to some extent spatial organisation, albeit in a “flattened” fashion.

2) The experiment testing for association between L1 larvae and proliferating centres lacks important information on the actual areas covered by proliferating and differentiated cells. This information is necessary to correctly interpret the numbers the authors provide. If the area of Ki67 positive cells is just 5% of the total area, this finding is highly specific. If 90% of the transwell-area is composed of Ki67 positive cells, even an association of 27/30 would merely indicate completely random distribution. According to the one overview picture the authors provide, the Ki67 positive area appears to be smaller compared to the differentiated one. Nevertheless, the authors should do proper statistics here that take the actual area of each region into account and tests for significance.

We agree with the reviewer that without extensive quantification of the Ki-67⁺ cells coverage of the organoid culture it is not possible to formally test an association between L1 larvae and the proliferating centres. However, the aim of this experiment was to show that, in addition to seeing the same cell types in the caecaloids as in the caecum, using the caecaloids the same parasite-cell type interactions are observed. Thus, even though occasionally worms are found that are not associated with Ki67⁺ cell types, the vast majority are clearly associated with proliferating (Ki-67⁺) cells. These qualitative observations provide evidence that the caecaloid system enables the study parasite-cell type interactions that are comparable to those that occur *in vivo*.

To clarify the aim of the experiment in the paper, we have rephrased the text so that no longer implies that a statistical association has been tested, line 162:

In fact, from a single experiment at 72h p.i., we recovered the majority of larvae (n=27/30) inside or in direct contact with clusters of Ki-67⁺ IECs (Fig. 3c).

References

- 1 Thorne, C. A. *et al.* Enteroid Monolayers Reveal an Autonomous WNT and BMP Circuit Controlling Intestinal Epithelial Growth and Organization. *Dev Cell* **44**, 624-633 e624, doi:10.1016/j.devcel.2018.01.024 (2018).

- 2 Wang, Y. *et al.* Long-Term Culture Captures Injury-Repair Cycles of Colonic Stem Cells. *Cell* **179**, 1144-1159 e1115, doi:10.1016/j.cell.2019.10.015 (2019).